# Critical spin chains and loop models with $PSU(n)$ symmetry

Paul Roux[1,2★], Jesper Lykke Jacobsen[1,3†], Sylvain Ribault[1‡] and Hubert Saleur[1,4∘]

**1** Institut de physique théorique, CEA, CNRS, Université Paris-Saclay
**2** Laboratoire de Physique de l'École Normale Supérieure, ENS, Université PSL, CNRS, Sorbonne Université, Université Paris Cité, F-75005 Paris, France
**3** Sorbonne Université, École Normale Supérieure, CNRS, Laboratoire de Physique (LPENS)
**4** Department of Physics and Astronomy, University of Southern California, Los Angeles

★ paul.roux@ens.fr , † jesper.jacobsen@ens.fr ,
‡ sylvain.ribault@ipht.fr , ∘ hubert.saleur@ipht.fr

## Abstract

Starting with the Ising model, statistical models with global symmetries provide fruitful approaches to interesting physical systems, for example percolation or polymers. These include the $O(n)$ model (symmetry group $O(n)$) and the Potts model (symmetry group $S_Q$). Both models make sense for $n, Q \in \mathbb{C}$ and not just $n, Q \in \mathbb{N}$, and both give rise to a conformal field theory in the critical limit. Here, we study similar models based on the group $PSU(n)$. We focus on the two-dimensional case, where the models can be described either as gases of non-intersecting orientable loops, or as alternating spin chains. This allows us to determine their spectra either by computing a twisted torus partition function, or by studying representations of the walled Brauer algebra. In the critical limit, our models give rise to a CFT that exists for any $n \in \mathbb{C}$ and has a global $PSU(n)$ symmetry. Its spectrum is similar to those of the $O(n)$ and Potts CFTs, but a bit simpler. We conjecture that the $O(n)$ CFT is a $\mathbb{Z}_2$ orbifold of the $PSU(n)$ CFT, where $\mathbb{Z}_2$ acts as complex conjugation.

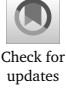

# 1  Introduction

**Statistical models with $PSU(n)$ and $O(n)$ global symmetries**

Statistical models of interacting spins are among the most popular statistical lattice models with a global symmetry. Their formulations as loop models provide a fruitful connection between statistical mechanics and random geometry. This is especially true in two dimensions, where a wealth of techniques are available, such as integrability, the conformal bootstrap, exact diagonalisation or matrix product states.

Good examples of this are the $O(n)$ and $PSU(n)$ models. These models can described in terms of spin variables of fixed norm $S_i \in \mathbb{R}^n$ or $Z_i \in \mathbb{C}^n$. The interaction energy between two nearest neighbour spins is

$$O(n): \quad \log\left(1 + x\, S_i S_j\right), \tag{1a}$$

$$PSU(n): \quad \log\left(1 + x\, Z_i^\dagger Z_j\right), \tag{1b}$$

where the parameter $x$ is called the interaction strength. These models can be reformulated for generic values of $n \in \mathbb{R}_+$ or even $n \in \mathbb{C}$ [1]. Admittedly, interesting physical examples tend to occur for $n \in \mathbb{N}$, for instance the dense $U(1)$ model describes percolation hulls. Nevertheless, it is interesting to study the models at generic $n$ as a unifying framework, which is also simpler algebraically than special cases. While the groups $O(n)$ or $PSU(n)$ a priori only make sense for $n \in \mathbb{N}$, there exists a generalization of group theory that describes global symmetries with a continuous parameter $n \in \mathbb{C}$ [2]. As we review in Section 2, this generalization is based on the tensorial structure of representations rather than elements of the group.

**Spin chains and loop models**

Statistical spin models can be described either as hamiltonian spin chains, or as lattice loop models.

In the spin chain, each site carries a representation of $O(n)$ or $U(n)$. For simplicity, we let each site carry either nothing or a spin. The crucial difference between the $O(n)$ and $U(n)$ cases is that $U(n)$ spins can belong to two different representations, the fundamental and the anti-fundamental, denoted $[1]$ and $\overline{[1]}$. Since in (1b) a spin always interacts with a complex conjugated spin, a simple possibility is to alternate $[1]$ and $\overline{[1]}$ with nearest-neighbour interactions:

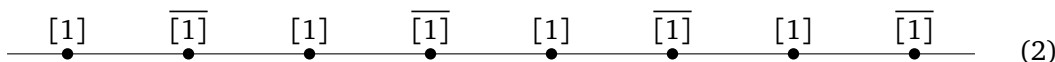

$$\tag{2}$$

The loop model description is obtained by evolving the spin chain along discrete Euclidean time, as we review in Section 3. The spin chain's dynamics give rise to configurations of loops. In the $U(n)$ case, since the interaction (1b) is asymmetric, loops come with an orientation, such that neighbouring loops have opposite orientations. On a torus in particular, this constrains the parity of the number of non-contractible loops.

Loop models based on $U(n)$ spins were previously studied in [3, 4], see also [5, 6]. A configuration of oriented loops on the square lattice is pictured below:

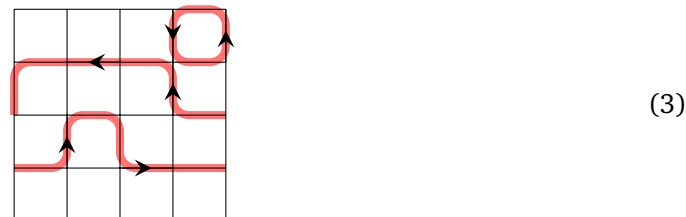

(3)

By high-temperature expansions on the lattice, the number $n$ of spin components becomes, in the loop model formulation, the weight of a loop. The interaction strength $x$ in eq. (1) becomes the weight of a monomer. For a large lattice, microscopic details such as the value of $x$, the geometry of the lattice, and the rules by which monomers can be drawn on a given vertex of the lattice have a limited impact on the physics at the critical point. Various choices for these details lead to few different universality classes. In contrast, the weight $n$ of loops is a global symmetry parameter, and a universal quantity.

**Global symmetry**

The alternated spin chain has a $PSU(n)$ global symmetry. Indeed, each site transforms in a representation of $U(n)$, but the global action of $U(1)$ is trivial, since $\rho \in U(1)$ acts on $[1] \otimes \overline{[1]}$ as $\rho\bar{\rho} = 1$. The chain thus has a global $PSU(n) = U(n)/U(1)$ symmetry. Correspondingly, the CFT spectrum (66) which we compute in section 4 has a current $V_{(1,-1)}$ of multiplicity $n^2 - 1 = \dim PSU(n)$. As we explain in detail in section 2, the representations of $U(n)$ that appear in the spectrum (66) of the $PSU(n)$ CFT are in bijection with representations of $PSU(n)$. Labelling them as $U(n)$ representations makes it more convenient to work at generic $n$. The $PSU(n)$ symmetry of the spin chain is consistent with the conjectured equivalence of the continuum limit with the $\mathbb{CP}^{n-1}$ model, whose Lagrangian is given in (152). This model has a $U(n)$ symmetry with a gauged $U(1)$ subgroup, leaving a global $PSU(n)$ global symmetry.

In early versions of this paper the model was named $U(n)$ model, after the spin model from which it is constructed. After discussion with the SciPost referees, we decided to name it $PSU(n)$ model to avoid confusion on the nature of its global symmetry. We had to modify the title of the paper in consequence. We do not exclude the possibility that there may exist generalisations of the model discussed in this paper which do have full $U(n)$ symmetry, though we could not find any.

**Phase diagrams**

The phase diagram of the $O(n)$ model is well-known, [7]. We plot it in figure (4). On the square lattice, the dense and dilute critical lines are $x^{-1} = \sqrt{2 \pm \sqrt{2-n}}$. To the best of our knowledge, the phase diagram of the $PSU(n)$ model was never studied. However, because of the close relation of the $PSU(n)$ model to the $O(n)$ and Potts models we expect this phase diagram to be similar to that of the $O(n)$ model in the $(x^{-1}, n)$ plane, though the exact equations

of the dense and dilute critical lines may change.

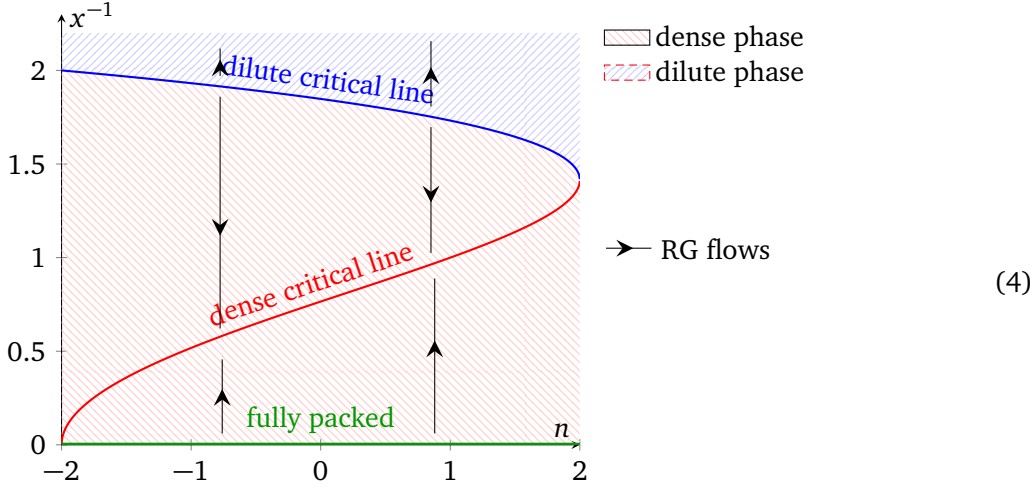

Phase diagram of the $O(n)$ and $PSU(n)$ loop models
(4)

For low values of $x$, the model is not critical, and consists of small, dilute loops. When $x$ reaches the critical value $x_c$, the loops become fractal. The corresponding dilute critical point is related with the standard critical point of the $\Phi^4$ Landau–Ginzburg universality class with $O(n)$ symmetry [3,8]. Increasing the monomer weight $x$ beyond the critical value takes the model into its so-called dense phase, which is also critical. We are not aware of an interpretation of the corresponding universality class in the Landau–Ginzburg language. It has recently been suggested that this phase can be analyzed using the language of non-invertible symmetries [9]. While it does look like a critical low temperature phase and is stable against changes of monomer weight, it is non generic from the $O(n)$ point of view, since crossings are now relevant and induce a flow towards the generic Goldstone phase $O(n)/O(n-1)$ where the symmetry is spontaneously broken [10]. (The Mermin–Wagner theorem does not apply due to the loss of unitarity when $n$ is not an integer.)

The dense and dilute critical phases of the $O(n)$ and $PSU(n)$ models are described by CFTs which we call the $O(n)$ and $PSU(n)$ CFTs. The central charges of the $O(n)$ and $PSU(n)$ CFTs are related to the parameter $n$ via [11]

$$n = -2\cos(\pi\beta^2),\tag{5}$$
$$c = 13 - 6(\beta^2 + \beta^{-2}).\tag{6}$$

For a given $n$, different determinations of $\beta$ describe the different phases: $\beta^2 \in (0,1)$ describes the dense phases, $\beta^2 \in [1,2]$ describes the dilute phases.

On the other hand, the fully packed line of the $O(n)$ loop model is described by a different CFT with central charge $c = c_{\text{dense}} + 1$ and an extended $W_3$ symmetry algebra [12], which comes from a $SU(3)_q$ symmetry of the loop model [7,13,14].

The spectra of the $O(n)$ and $PSU(n)$ CFTs are representations of the conformal algebra $\mathfrak{C}_c = \text{Vir}_c \oplus \overline{\text{Vir}}_c$, where $\text{Vir}_c$ is the Virasoro algebra of central charge $c$. They are built from the same kinds of primary fields, which fall into the following two classes:

| Field | Indices | Name | conformal dimensions | Conformal module |
|---|---|---|---|---|
| $V^d_{(1,s)}$ | $s \in \mathbb{N}^*$ | Degenerate | $(\Delta_{(1,s)}, \Delta_{(1,s)})$ | $\mathscr{R}_{\langle 1,s\rangle}$ |
| $V_{(r,s)}$ | $rs \in \mathbb{Z}$ | Non-diagonal | $(\Delta_{(r,s)}, \Delta_{(r,-s)})$ | $\mathscr{W}_{(r,s)}$ |

(7)

where

$$\Delta_{(r,s)} = p_{(r,s)}^2 - p_{(1,1)}^2, \quad \text{with} \quad p_{(r,s)} := \frac{1}{2}(\beta r - \beta^{-1}s), \tag{8}$$

and we indicate the left and right conformal dimensions. In these notations, the sets of primary fields of the $O(n)$ and $PSU(n)$ CFTs are

$$\mathcal{S}^{O(n)} = \left\{ V_{\langle 1,s \rangle}^d \right\}_{s \in 2\mathbb{N}+1} \bigcup \left\{ V_{(r,s)} \right\}_{\substack{r \in \frac{1}{2}\mathbb{N}^* \\ s \in \frac{1}{r}\mathbb{Z}}}, \tag{9}$$

$$\mathcal{S}^{PSU(n)} = \left\{ V_{\langle 1,s \rangle}^d \right\}_{s \in \mathbb{N}^*} \bigcup \left\{ V_{(r,s)} \right\}_{\substack{r \in \mathbb{N}^* \\ s \in \frac{1}{r}\mathbb{Z}}}. \tag{10}$$

The spectra are however not completely characterized by the primary fields and their conformal dimensions. This is first because non-diagonal fields $V_{(r,s)}$ with $r \in \mathbb{N}^*$ and $s \in \mathbb{Z}^*$ actually belong to logarithmic representations, whose structure was elucidated in [15]. Most importantly for us, this is because fields are also characterized by how they transform under the global symmetry group. We will derive the representations of $PSU(n)$ in which our fields transform, see Eq. (66) for the result. Comparable results for the $O(n)$ and Potts CFTs were derived in [16] and [17].

In physical terms, the degenerate field $V_{\langle 1,3 \rangle}^d$ is called a local energy operator. A perturbation in $V_{\langle 1,3 \rangle}^d$ changes the weight of monomers. In particular, the RG flows of the phase diagram (4) are generated by $V_{\langle 1,3 \rangle}^d$, which is relevant on the dilute critical line and irrelevant on the dense critical line, since $\Delta_{(1,3)} - 1 = 2(\beta^{-2} - 1)$. The $PSU(n)$ CFT can be perturbed by $V_{\langle 1,2 \rangle}^d$, which is relevant if and only if $n > 0$, since $\Delta_{\langle 1,2 \rangle} - 1 = \frac{3}{4}(\beta^{-2} - 2)$. The corresponding RG flow introduces a new coupling, taking the model out of the $(x^{-1}, n)$ plane. The higher degenerate fields are responsible for corrections to the scaling of $V_{\langle 1,2 \rangle}^d, V_{\langle 1,3 \rangle}^d$. Non-diagonal fields correspond in the statistical model to insertions of spins. In particular, the non-diagonal field $V_{(2,0)}$ describes the crossing of loops. This field is relevant on the dense critical line and irrelevant on the dilute critical line, since $\Delta_{(2,0)} - 1 = \frac{1}{4}(3 + \beta^{-2})(\beta^2 - 1)$.

**Orbifold relation**

The similarity between the spectra of the $O(n)$ and $PSU(n)$ CFT spectra suggests these CFTs are closely related. In Section 5 we find evidence that the $O(n)$ CFT is a $\mathbb{Z}_2$ orbifold of the $PSU(n)$ CFT. As a first hint, note that the $O(n)$ CFT's spectrum has non-diagonal fields with $r \in \mathbb{N} + \frac{1}{2}$, in addition to the $PSU(n)$ CFT's $r \in \mathbb{N}^*$. We will interpret these extra non-diagonal fields as the orbifold's twisted sector.

# 2 Representation theory of $U(n)$

In this section we review the complex irreducible representations of $U(n)$: their structure, dimensions, tensor products, and decompositions into $O(n)$ representations. These features are implemented in a publicly available code [18]. We also review how diagram algebras arise when studying the category of representations of $U(n)$, and compare with the representation theory of $O(n)$. Some of these facts are summarized in the Wikipedia article Representations of classical Lie groups. In a nutshell, we will review the following objects:

| Group | $U(n)$ | $O(n)$ |
|---|---|---|
| Representations | Mixed-tensorial: $\subset [1]^{\otimes p} \otimes \overline{[1]}^{\otimes q}$ | Tensorial: $\subset [1]^{\otimes p}$ |
| Irreps set | Young bidiagrams $\lambda\bar{\mu}$ | Young diagrams $\lambda$ |
| Structure constants | $\Gamma^{\nu\bar{\rho}}_{\lambda_1\bar{\mu}_1,\lambda_2\bar{\mu}_2}$ (18) | Newell-Littlewood numbers |
| Diagram algebra | walled Brauer $\mathscr{B}_{p,q}(n)$ | Brauer $\mathscr{B}_p(n)$ |

## 2.1 Tensors and representations of $U(n)$

The group $U(n)$ of complex matrices $A$ such that $A^\dagger A = 1$ is a real Lie group of dimension $n^2$. It is the Lie compact real form of $GL(n)$, which implies that complex linear representations of the two groups are in one-to-one correspondence via the inclusion $U(n) \hookrightarrow GL(n)$. We write the coefficient-by-coefficient complex conjugation of $U(n)$ as

$$c\colon U(n) \to U(n), \quad g = (g_{ij}) \mapsto \bar{g} = (\bar{g}_{ij}), \tag{11}$$

whereas the adjoint map $g \mapsto g^\dagger$ acts as $(g^\dagger)_{ij} = (\bar{g}^T)_{ij} = \bar{g}_{ji}$. Because $c$ is a group morphism, given a representation $(R, \rho)$ of $U(n)$, $(R, \rho \circ c)$ is also a representation of $U(n)$. Since for $g \in U(n)$, $c(g) := \bar{g} = (g^{-1})^T$, $(R, \rho \circ c)$ is nothing but the representation dual (or contragredient) to $R$, which we henceforth denote $\bar{R}$.

**Irreducible representations**

The group $U(n)$ has an obvious irreducible representation: the fundamental representation $V = \mathbb{C}^n$, where a matrix $g \in U(n)$ acts by matrix multiplication $g \cdot v = gv$. The representation $\bar{V}$ dual to $V$ is also irreducible, and is not isomorphic to $V$. It is called the antifundamental representation.

The fundamental theorem about finite-dimensional representations of $U(n)$ is that they are all mixed-tensorial, i.e. subrepresentations of a given $V^{\otimes p} \otimes (\bar{V})^{\otimes q}$ for some $p, q \in \mathbb{N}$. For proofs of the following results we refer the reader to [19, paragraph 15.5] or [20].

Irreducible representations of $U(n)$ are indexed by Young bidiagrams of length $\leq n$. A Young bidiagram, also called a bipartition, is a pair of Young diagrams. To lighten notations we use exponents for repeated integers; for instance $[3221] = [32^2 1]$ is the diagram

$$[32^2 1] \;\; = \;\; \begin{array}{l} \square\square\square \\ \square\square \\ \square\square \\ \square \end{array} \tag{12}$$

The size $|\lambda|$ of a diagram $\lambda$ is its number of boxes, its length $\ell(\lambda)$ is its number of rows, and we call $\lambda_i$ the number of boxes in its $i$-th row. A Young bidiagram is a pair of Young diagrams $(\lambda, \mu)$, which we will write as $\lambda\bar{\mu}$. When at least one of the diagrams is the empty Young diagram $[\,]$, we generally use the more concise notations $\lambda = \lambda[\bar{\,}]$ or $\bar{\mu} = [\,]\bar{\mu}$ or $[\,] = [\,][\bar{\,}]$. The size and length of a bidiagram are defined as $|\lambda\bar{\mu}| := |\lambda| + |\mu|$, $\ell(\lambda\bar{\mu}) := \ell(\lambda) + \ell(\mu)$. The transpose $\tilde{\lambda}$ of a diagram is the diagram such that $\tilde{\lambda}_i$ is the number of boxes in the $i$-th column of $\lambda$.

Loosely speaking, the irreducible representation $\lambda\bar{\mu}$ is the traceless subspace of $\mathbb{S}^\lambda V \otimes \mathbb{S}^\mu(\bar{V})$, where $\mathbb{S}^\lambda$ is a Schur functor, and by traceless we mean that any contraction of a covariant index (from $V$) with a contravariant one (from $\bar{V}$) yields zero. As for the Schur functor, we will content ourselves with saying that it symmetrizes according to the rows

of the corresponding diagram, and anti-symmetrizes according to its columns. For instance $\mathbb{S}^{[k]}V = \mathrm{Sym}^k(V)$, $\mathbb{S}^{[1^k]}V = \bigwedge^k(V)$. In particular, $\overline{\lambda\bar{\mu}} = \mu\bar{\lambda}$, hence our notation for bipartitions. Note also that the action of $U(1) \subset U(n)$ on $\lambda\bar{\mu}$ is trivial if $|\lambda| = |\mu|$.

In these notations, $V = [1]$, $\bar{V} = \overline{[1]}$. We now discuss two examples. First, $[1]\overline{[1]}$ is made of tensors $A^i{}_j$ with one covariant and one contravariant index, such that $A^i{}_i = 0$, so

$$\dim[1]\overline{[1]} = n^2 - 1, \quad n \geq 2. \tag{13}$$

This is the dimension of the adjoint representation of $SU(n)$. Indeed, since $\mathfrak{u}(n) = \mathfrak{su}(n) \oplus \mathfrak{u}(1)$, the adjoint representation of $U(n)$ is not irreducible, and $[1]\overline{[1]}$ is the non-trivial irreducible part of the adjoint of $U(n)$. Second, $[1^2]\overline{[1^2]}$ is made of tensors $A^{ij}{}_{kl}$ antisymmetric in both pairs of indices and such that $A^{ij}{}_{il} = 0$ (using Einstein summation conventions) which yields $n^2$ independent relations. In particular it has dimension

$$\dim[1^2]\overline{[1^2]} = \left(\frac{n(n-1)}{2}\right)^2 - n^2 = \frac{n^2(n+1)(n-3)}{4}\,v\,. \tag{14}$$

The dimension of $\lambda\bar{\mu}$ is given by the combinatorial formula [21]

$$\dim \lambda\bar{\mu} = \prod_{(i,j)\in\lambda} \frac{n-1+i+j-\tilde{\lambda}_j-\tilde{\mu}_i}{h_\lambda(i,j)} \prod_{(i,j)\in\mu} \frac{n+1-i-j+\lambda_j+\mu_i}{h_\mu(i,j)}\,, \tag{15}$$

where $(i,j)$ is the box of the diagram at the $i$-th row and $j$-th column, and $h_\lambda(i,j)$ is the hook length at $(i,j)$, i.e. the number of boxes to the right or below $(i,j)$, including $(i,j)$ itself. Below are a few examples:

$$\dim[1]\overline{[1]} = n^2 - 1\,, \tag{16a}$$

$$\dim[2]\overline{[2]} = \frac{n^2(n+3)(n-1)}{4}\,, \tag{16b}$$

$$\dim[3]\overline{[3]} = \frac{(n+5)(n+1)^2(n-1)n^2}{36}\,, \tag{16c}$$

$$\dim[3]\overline{[21]} = \frac{n^2(n-2)(n+2)^2(n+4)}{18}\,. \tag{16d}$$

The character of a representation $R$ of $U(n)$ is

$$g \mapsto \chi_R(g) = \mathrm{Tr}_R\, g\,. \tag{17}$$

This is a symmetric polynomial in the eigenvalues $x_1, \ldots, x_n$ of $g$, and in their complex conjugates.

**Tensor product decomposition**

The decomposition rules for tensor products of irreducible representations of $U(n)$ are known [20]. Whenever $|\lambda_1\bar{\mu}_1| + |\lambda_2\bar{\mu}_2| \leq n$,

$$
\begin{aligned}
\lambda_1\bar{\mu}_1 \otimes \lambda_2\bar{\mu}_2 &= \sum_{\nu\bar{\rho}} \Gamma^{\nu\bar{\rho}}_{\lambda_1\bar{\mu}_1,\lambda_2\bar{\mu}_2}\, \nu\bar{\rho}\,, \\
\Gamma^{\nu\bar{\rho}}_{\lambda_1\bar{\mu}_1,\lambda_2\bar{\mu}_2} &= \sum_{\alpha,\beta,\eta,\theta} \left(\sum_\kappa c^{\lambda_1}_{\kappa,\alpha} c^{\mu_2}_{\kappa,\beta}\right)\left(\sum_\gamma c^{\lambda_2}_{\gamma,\eta} c^{\mu_1}_{\gamma,\theta}\right) c^\nu_{\alpha,\theta} c^\rho_{\beta,\eta}\,,
\end{aligned}
\tag{18}
$$

where all indices range over the whole set of Young diagrams. The coefficients $c^\nu_{\lambda,\mu} \in \mathbb{N}$ are Littlewood-Richardson coefficients. The coefficient $c^\lambda_{\mu,\nu}$ vanishes when $\mu$ or $\nu$ get large, which

ensures the sums are finite. The tensor product of pure-tensorial representations is obtained as a particular case:

$$\lambda \otimes \mu = \sum_{\delta} c^{\delta}_{\lambda,\mu} \delta \,. \tag{19}$$

Below are a few examples of applications of this formula:

$$[1] \otimes [1] = [2] + [1^2], \tag{20a}$$

$$[1] \otimes \overline{[1]} = [1]\overline{[1]} + [\,], \tag{20b}$$

$$[1] \otimes [1]\overline{[1]} = [2]\overline{[1]} + [1^2]\overline{[1]} + [1], \tag{20c}$$

$$[1]\overline{[1]} \otimes [1]\overline{[1]} = [2]\overline{[2]} + [2]\overline{[11]} + [11]\overline{[2]} + [11]\overline{[11]} + 2[1]\overline{[1]} + [\,], \tag{20d}$$

$$[k] \otimes \overline{[k]} = \sum_{i=0}^{k} [i]\overline{[i]}, \tag{20e}$$

$$[4] \otimes \overline{[21^2]} = [4]\overline{[21^2]} + [3]\overline{[21]} + [3]\overline{[1^3]} + [2]\overline{[1^2]}. \tag{20f}$$

**Branching to $O(n)$**

Because $O(n)$ is a subgroup of $U(n)$, it is possible to decompose irreducible representations of $U(n)$ under the action of $O(n)$. The branching rules from $U(n) \downarrow O(n)$ are given by [22]

$$\lambda \bar{\mu} \underset{O(n)}{=} \sum_{\nu} m^{\lambda\bar{\mu}}_{\nu} \nu^{O(n)}, \quad m^{\lambda\bar{\mu}}_{\nu} = \sum_{\alpha,\beta,\gamma,\delta} c^{\nu}_{\alpha,\beta} c^{\lambda}_{\alpha,2\gamma} c^{\mu}_{\beta,2\delta}, \tag{21}$$

where $\nu^{O(n)}$ is the representation of $O(n)$ labelled by the Young diagram $\nu$ [17], $c^{\lambda}_{\mu,\nu}$ are again Littlewood-Richardson coefficients, $2\lambda$ is the diagram such that $(2\lambda)_i = 2\lambda_i$, and the sums range over the set of all Young diagrams. We often omit the superscript in $\nu = \nu^{O(n)}$, when there is no risk of confusion with the $U(n)$ representation $\nu = \nu[\bar{\,}]$. In the case of pure-tensorial representations of $U(n)$, the branching rules reduce to

$$\lambda \underset{O(n)}{=} \sum_{\nu,\gamma} c^{\lambda}_{\nu,2\gamma} \nu^{O(n)} \,. \tag{22}$$

Below are a few examples of branchings $U(n) \downarrow O(n)$:

$$[k]\overline{[\,]} \underset{O(n)}{=} \sum_{i=0}^{\lfloor \frac{k}{2} \rfloor} [k-2i], \tag{23a}$$

$$[1]\overline{[1]} \underset{O(n)}{=} [2] + [1^2], \tag{23b}$$

$$[4]\overline{[21]} \underset{O(n)}{=} [61] + [52] + [51^2] + [421] + [5] + 2[41] + [32] + [31^2]$$
$$+ [2^2 1] + [3] + 2[21] + [1]. \tag{23c}$$

The term $2[21]$ means two copies of the representation $[2]$, not to be confused with the previous notation for $2\lambda$. Equation (23b) was to be expected since the adjoint of $U(n)$ is made of anti-hermitian matrices, which decompose into $[2]$, symmetric traceless matrices, and $[1^2]$, antisymmetric matrices.

**Representations of $SU(n)$**

Irreducible representations of $SU(n)$ are indexed by Young diagrams $\nu$ with $\ell(\nu) < n$, and we denote them as $\nu^{SU(n)}$. Let us see how these relate to the representations $\lambda\bar{\mu}$. Since

$SU(n) \subset U(n)$, any representation $\rho : U(n) \to \lambda\bar{\mu}$ can be restricted to a representation of $SU(n)$. When $\ell(\nu) < n$, the representation $\nu\overline{[\ ]}$ of $U(n)$ restricts to the irreducible representation $\nu^{SU(n)}$ [19, paragraph 15.5]. More generally, since $U(n)/SU(n) = U(1)$, the restriction of any irreducible representation of $U(n)$ is an irreducible representation of $SU(n)$. To see which one it is, let us introduce a new labeling for irreducible representations of $U(n)$:

$$\lambda\bar{\mu} = V_{\lambda_1,\dots,\lambda_k,\underbrace{0,\dots,0}_{n-k-k'},-\mu_{k'},\dots,-\mu_1}, \tag{24}$$

where $\lambda = [\lambda_1 \dots \lambda_k]$ and $\mu = [\mu_1 \dots \mu_{k'}]$. This is convenient for writing the tensor product with the determinant representation $D = [1^n]$ of $U(n)$ and its conjugate $D^{-1} = \overline{[1^n]}$ [19, paragraph 15.5]:

$$V_{a_1,\dots,a_n} \otimes D^{\otimes r} = V_{a_1+r,\dots,a_n+r}. \tag{25}$$

In terms of Young diagrams, this translates to

$$\lambda\bar{\mu} \otimes D^{\otimes r} = [(\lambda_1+r)\dots(\lambda_k+r)\underbrace{r\dots r}_{n-k-k'}(r-\mu_{k'})\dots(r-\mu_2)(r-\mu_1)]. \tag{26}$$

Pictorially, tensoring with $D$ amounts to removing the first column of length $\tilde{\mu}_1$ from $\mu$ and adding a column of length $n - \tilde{\mu}_1$ to $\lambda$; below is an example, drawn for $U(6)$:

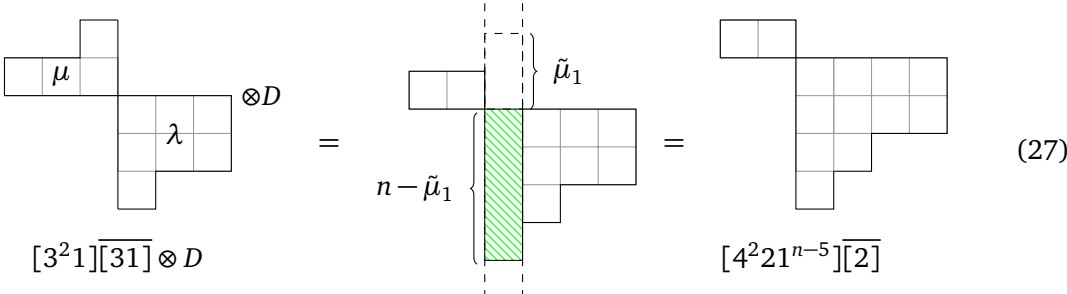

$$\tag{27}$$

Similarly, tensoring with $D^{-1}$ removes the first column $\tilde{\lambda}_1$ from $\lambda$ and adds a column of length $n - \tilde{\lambda}_1$ to $\mu$.

Iterating this procedure, we now show that one can always reach a pair of diagrams of the form $\nu\overline{[\ ]}$ with $\ell(\nu) \leq n-1$, which restricts to the irreducible representation $\nu^{SU(n)}$ of $SU(n)$. There are two cases to consider. First, if $\mu \neq [\ ]$, the RHS of (26) is a Young diagram with at most $n-1$ lines, thus it corresponds to an irreducible representation of $SU(n)$. If on the other hand $\mu = [\ ]$ and $\lambda$ has $n$ lines, tensoring with $D^{-1}$ removes the first column of $\lambda$. Iterating, one can again reach a Young diagram with at most $n-1$ lines. In the above example for instance we get

$$[3^2 1]\overline{[31]} \otimes D^{\otimes 3} = [6^2 4 3^{n-5} 2]\overline{[\ ]} \underset{SU(n)}{=} [6^2 4 3^{n-5} 2]^{SU(n)}. \tag{28}$$

### Representations of $PSU(n)$

On a representation $\lambda\bar{\mu} \subset [1]^{\otimes|\lambda|} \otimes \overline{[1]}^{\otimes|\mu|}$, $\rho \in U(1)$ acts diagonally as $\rho^{|\lambda|-|\mu|}$. We denote $\mathrm{Rep}_0\, U(n)$ the set of representations of $U(n)$ on which $U(1)$ acts trivially, which are direct sums of irreducible representations with $|\lambda| = |\mu|$:

$$\mathrm{Rep}_0\, U(n) = \mathrm{Span}_{\mathbb{N}}\{\lambda\bar{\mu}, |\lambda| = |\mu|\}. \tag{29}$$

By the universal property of the quotient, representations in $\mathrm{Rep}_0 U(n)$ descend to representations of the quotient $PSU(n) = U(n)/U(1) = SU(n)/\mathbb{Z}_n$:

$$
\begin{array}{ccc}
U(n) & \xrightarrow{\quad\rho\quad} & \mathrm{End}(\lambda\bar{\mu}) \\
\downarrow{\scriptstyle\pi} & \nearrow{\scriptstyle\exists!\,\tilde{\rho}} & \\
SU(n) & \longrightarrow & PSU(n)
\end{array}
\tag{30}
$$

This gives an injective map from $\mathrm{Rep}_0 U(n)$ to a subset of the representations of $PSU(n)$ or $SU(n)$. We also call $\mathrm{Rep}_0 PSU(n)$ or $\mathrm{Rep}_0 SU(n)$ the image of this map, which is the following subset of all representations of $PSU(n)$:

$$
\mathrm{Rep}_0 PSU(n) \simeq \mathrm{Rep}_0 SU(n) = \{\nu^{SU(n)}, n \mid |\nu|\}.
\tag{31}
$$

The inclusion from left to right is obtained by tensoring with $r = \mu_1$ determinants in the formula (26):

$$
\lambda\bar{\mu} \otimes D^{\otimes\mu_1} = [(\lambda_1 + \mu_1)\ldots(\lambda_k + \mu_1)\underbrace{\mu_1\ldots\mu_1}_{n-k-k'}(\mu_1 - \mu_{k'})\ldots(\mu_1 - \mu_2)0].
\tag{32}
$$

The RHS has length $|\lambda| + (n-1)\mu_1 - (|\mu| - \mu_1) = n\mu_1$ which is indeed a multiple of $n$. Conversely, if $|\nu| = rn$, equation (26) gives

$$
\nu\overline{[\ ]} \otimes D^{-r} = [(\nu_1 - r)\ldots(\nu_k - r)\underbrace{(-r)\ldots(-r)}_{n-k}].
\tag{33}
$$

The sum of the coefficients on the RHS is $|\nu| - nr = 0$, and since the Young diagrams $\lambda, \mu$ corresponding to these coefficients are respectively given by the positive and negative coefficients, they verify $|\lambda| = |\mu|$.

## 2.2 Walled Brauer algebra and representations of $U(n)$

The Lie group $PSU(n)$ is not quite the right object to describe the symmetry of the $PSU(n)$ model. In general, in quantum field theory, we need only know two things about a symmetry group:

- its representations, in which the states transform, and

- their tensor products, which constrain interactions.

For instance, to build a Lagrangian bilinear in fields $\psi, \varphi$ transforming in representations $R, R'$ of a given group, one needs to know the projector $R \otimes R' \to [\ ]$ where $[\ ]$ is the trivial representation of the group. Similarly to construct a correlator out of fields transforming in representations $R_1 \ldots R_n$, one needs to know the projector $R_1 \otimes \cdots \otimes R_n \to [\ ]$.

This means that we are not so much interested in the representations themselves, but rather in the structure of their tensor products, and in morphisms between representations. Formally, this structure is encoded by the category of representations $\mathrm{Rep}\, PSU(n)$, whose objects are representations of $PSU(n)$ and whose morphisms are equivariant maps (also called intertwiners). And we want to focus on the $n$-independent part of the structure of $\mathrm{Rep}\, PSU(n)$. The correct setting for this is the formalism of Deligne categories.

There are four Deligne categories associated to series of classical groups: $\underline{\mathrm{Rep}}\, S_Q$, $\underline{\mathrm{Rep}}\, O(n)$, $\underline{\mathrm{Rep}}\, Sp(2n)$ and $\underline{\mathrm{Rep}}\, U(n)$. As a tensor category, $\underline{\mathrm{Rep}}\, Sp(2n)$ is equivalent to $\underline{\mathrm{Rep}}\, O(-2n)$, thus leaving three inequivalent Deligne categories. The Potts and $O(n)$ CFTs are CFTs whose global symmetries are described by $\underline{\mathrm{Rep}}\, S_Q$ and $\underline{\mathrm{Rep}}\, O(n)$. In Section 4, we will show that the closely related $PSU(n)$ model is a CFT with a global symmetry described by $\underline{\mathrm{Rep}}\, U(n)$, thus completing the picture. More precisely, we show that the model's symmetry is described by the category $\underline{\mathrm{Rep}}_0 PSU(n)$ which is isomorphic to a subcategory of $\underline{\mathrm{Rep}}\, U(n)$.

**Objects**

Take for instance the representation $[1^3]$ of $U(n)$. For any $n \in \mathbb{N}$, it is made of antisymmetric 3-tensors whose indices can take $n$ values. For any $n \geq 3$, this is an irreducible representation of $U(n)$ of dimension $\frac{n(n-1)(n-2)}{6}$ according to (15). Accidents occur for $n = 1, 2$, where this representation is 0-dimensional. The Deligne category $\underline{\mathrm{Rep}}\, U(n)$ forgets about these accidents and for any $n \in \mathbb{C}$ it contains an object $[1^3]$ of dimension $\frac{n(n-1)(n-2)}{6}$. A similar phenomenon occurs in tensor products: for any $n \geq 3$,

$$[1] \otimes [1^2] = [21] + [1^3]. \tag{34}$$

Because $[1^3] = 0$ as a representation of $U(2)$, the tensor product becomes in this case

$$[1] \otimes [1^2] \underset{U(2)}{=} [21]. \tag{35}$$

Again, $\underline{\mathrm{Rep}}\, U(n)$ ignores these subtleties, and the tensor product of the objects $[1]$ and $[1^2]$ of $\underline{\mathrm{Rep}}\, U(n)$ is given by the $n$-independent formula (34).

**Characters**

There also exists a notion of character in the Deligne category, called universal characters in [20]; they are defined as the projective limit of characters of representations of $U(n)$. In words, they are objects $\chi_{\lambda\bar\mu}$ which, when evaluated on $n$ variables, give back the usual characters $\chi_{\lambda\bar\mu}$ of $U(n)$, provided $\ell(\lambda) + \ell(\mu) \leq n$.

In the universal character ring, one can substract a character from another, or multiply it by a complex coefficient. By extension we call *formal representation* a linear combination of irreducible representations whose coefficients are not all non-negative integers.

**Morphisms**

Recall that any representation of $U(n)$ is a subrepresentation of some $V_{p,q} := [1]^{\otimes p} \otimes \overline{[1]}^{\otimes q}$. In this short review we will only describe morphisms $V_{p,q} \to V_{p',q'}$, following [20, lemma 1.2]. General morphisms in $\underline{\mathrm{Rep}}\, U(n)$ are deduced from these by the categorical construction of $\underline{\mathrm{Rep}}\, U(n)$, see [2].

Let us start with the case $p = p', q = q'$. The action of $U(n)$ commutes with the action of $S_p \times S_q$, where $S_p$ (resp. $S_q$) is the symmetric group that permutes factors of $[1]$ (resp. $\overline{[1]}$). Said otherwise, elements $S_p \times S_q$ are equivariant morphisms. Moreover, the action of $U(n)$ on a factor $[1] \otimes \overline{[1]}$ commutes with the contraction map $c_{ij}: V_{p,q} \to V_{p-1,q-1}$

$$c_{ij}: v_1 \otimes \cdots v_i \cdots \otimes v_p \otimes \bar v_1 \otimes \cdots \bar v_j \cdots \otimes \bar v_q \mapsto \langle v_i, \bar v_j \rangle v_1 \otimes \cdots \hat v_i \cdots \otimes v_p \otimes \bar v_1 \otimes \cdots \hat{\bar v}_j \cdots \otimes \bar v_q, \tag{36}$$

where $\langle\ ,\ \rangle$ is the canonical pairing between $[1]$ and $\overline{[1]}$ and $\hat v_i$ means the factor is omitted. Lastly, the action of $U(n)$ also commutes with the map $d_{i,j}: V_{p-1,q-1} \to V_{p,q}$, which we now define in terms of a basis $(f_a)_{a=1,\cdots,n}$ of $[1]$ and its dual $(\bar f_a)$:

$$d_{i,j}: v_1 \otimes \cdots \otimes v_{p-1} \otimes \bar v_1 \otimes \cdots \otimes \bar v_{q-1} \mapsto \sum_a v_1 \otimes \cdots \underbrace{f_a}_{i^{\text{th}}} \cdots \otimes v_{p-1} \otimes \bar v_1 \otimes \cdots \underbrace{\bar f_a}_{j^{\text{th}}} \cdots \otimes \bar v_{q-1}. \tag{37}$$

There exists a nice graphical representation of these operators: let us picture the space $[1]^{\otimes p} \otimes \overline{[1]}^{\otimes q}$ as

$$
\underbrace{\uparrow \cdots \uparrow}_{p \text{ sites}} \underbrace{\downarrow \cdots \downarrow}_{q \text{ sites}} . \tag{38}
$$

We can then represent an element of $S_p \times S_q$ as an *oriented diagram*, as in the following example with $((124),(12)) \in S_4 \times S_2$

$$
\cdots = ((124),(12)). \tag{39}
$$

The morphisms $c_{ij}$ and $d_{i,j}$ can also be represented as oriented diagrams:

$$
\cdots = c_{3,2} \colon V_{4,2} \to V_{3,1}, \tag{40}
$$

$$
\cdots = d_{4,1} \colon V_{3,1} \to V_{4,2}. \tag{41}
$$

Note that all lines respect the arrow orientations; this is in fact why we have chosen to represent $[1]$ and $\overline{[1]}$ as up and down arrows. Moreover, the composition of morphisms can be represented graphically by stacking the diagram from top to bottom when reading the product from left to right, up to replacing loops disconnected from the outer sites by factors of $n$. We call this stacking operation the multiplication of diagrams. It is through these factors of $n$ that the Deligne category still depends on $n$.

As is shown in [20], all equivariant morphisms $V_{p,q} \to V_{p',q'}$ can be obtained from permutations, $c_{i,j}$'s and $d_{i,j}$'s by sum and composition. Graphically, this means that all morphisms can be represented as diagrams between rows of $p, q$ and $p', q'$ up and down sites, such that each site is linked to exactly one other site and such that the links respect the orientations of arrows. This whole diagrammatic representation for the category Rep $U(n)$ can be extended to other Lie groups such as $SU(n)$, $O(n)$, $SO(n)$, $Sp(2n)$, exceptional Lie groups, etc. It is sometimes called the birdtrack formalism [23].

**The walled Brauer algebra**

In the particular case $p = p', q = q'$, the diagrams along with their multiplication rule define an algebra, called the *walled Brauer algebra* $\mathscr{B}_{p,q}(n)$ [24]. As an example, below are two walled Brauer diagrams, and the result of their multiplication:

$$
\cdots \times \cdots = \cdots = n \cdot \cdots \tag{42}
$$

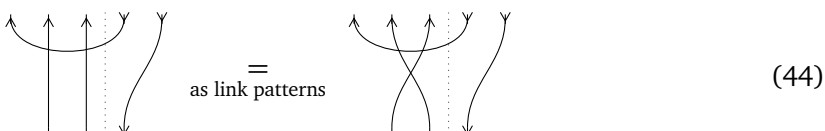

$$\text{(as link patterns)} \qquad (44)$$

The walled Brauer algebra $\mathscr{B}_{p,q}(n)$ is easily seen to have dimension $(p+q)!$. Indeed, upon reflecting all sites on the right side of the wall along a horizontal line, any walled Brauer diagram turns into a permutation of the $p+q$ sites. For $n \notin \mathbb{Z}$, $\mathscr{B}_{p,q}(n)$ is semisimple [25]. Its simple modules (irreducible representations) $B_{p,q}^{\lambda\bar{\mu}}$ are indexed by Young bidiagrams $\lambda\overline{\mu}$, where

$$|\lambda| \le p, \quad |\mu| \le q, \quad p - |\lambda| = q - |\mu|. \qquad (43)$$

The simple modules $B_{p,q}^{\lambda\bar{\mu}}$ can be described in terms of link patterns, which are diagrams with a top row of $p$ up and $q$ down sites, where two diagrams are identified if they only differ by a permutation of their bottom sites. The bottom sites are called defects.[1] For instance, the following diagrams correspond to the same link pattern:

Let us define

$$d_{|\lambda|,|\mu|}^{p,q} = \left\{ \text{link patterns with } |\lambda|, |\mu| \text{ up and down defects and } p, q \text{ sites} \right\}, \qquad (45)$$

and temporarily use the notations $\lambda, \mu$ for Specht modules of the symmetric group $S_{|\lambda|}$, then irreducible modules of the walled Brauer algebra are

$$B_{p,q}^{\lambda\bar{\mu}} = \mathbb{C} d_{|\lambda|,|\mu|}^{p,q} \otimes_{S_{|\lambda|} \times S_{|\mu|}} (\lambda \otimes \mu). \qquad (46)$$

In (46), $\lambda$ and $\mu$ are Specht modules, which are the irreducible modules of the symmetric group, indexed by Young diagrams. The notation $\otimes_{S_{|\lambda|} \times S_{|\mu|}}$ means that the action of a walled Brauer diagram on an element in $B_{p,q}^{\lambda\bar{\mu}}$ is obtained by

- multiplying the link pattern by the diagram, following the diagrammatic rules

- if the diagram multiplication has caused up or down defects to be permuted, letting the corresponding permutations act on the element of $\lambda$ and $\mu$.

For instance, in $B_{3,1}^{[1^2]}$,

$$\qquad \cdot \qquad = - \qquad \qquad (47)$$

In particular, $B_{p,q}^{\lambda\bar{\mu}}$ has dimension

$$\dim B_{p,q}^{\lambda\bar{\mu}} = \binom{p}{|\lambda|}\binom{q}{|\mu|}(p-|\lambda|)! \dim\lambda \dim\mu, \qquad (48)$$

Notice that this expression is symmetric under $p \leftrightarrow q, \lambda \leftrightarrow \mu$, thanks to Eq. (43).

---

[1]One should not confuse this notion of defects with that of defects in field theory

**The category $\underline{\mathrm{Rep}}_0 PSU(n)$**

It is not possible to construct Deligne categories for $SU(n)$ or $PSU(n)$ using a similar procedure as for $\underline{\mathrm{Rep}}\, U(n)$, see [2, footnote 38]. Indeed, the whole point of the Deligne category is to forget about small-$n$ accidents as in (34, 35), but the identification $[1^n] = [\,]$ is an $n$-dependent condition. And indeed there is no known Deligne category for $SU(n)$ nor $PSU(n)$.

However since $\mathrm{Rep}_0 U(n)$ is stable under tensor product, one can define a corresponding category $\underline{\mathrm{Rep}}_0 U(n) \subsetneq \underline{\mathrm{Rep}}\, U(n)$. Because $\mathrm{Rep}_0 U(n)$ is in one-to-one mapping with $\mathrm{Rep}_0 PSU(n)$, the category $\underline{\mathrm{Rep}}_0 U(n)$ can be interpreted as a Deligne category for $\mathrm{Rep}_0 PSU(n)$. For instance, under this mapping the adjoint representation $[21^{n-2}]$ of $PSU(n)$ is mapped to $[1]\overline{[1]}$ independently of $n$, since $[1]\overline{[1]} \otimes D = [21^{n-2}]$ according to equation (26). This allows to make sense of tensor products of representations in $\mathrm{Rep}_0 PSU(n)$ independently of $n$.

On the contrary, there is no canonical way to associate a representation of $U(n)$ to a representation which is not in $\mathrm{Rep}_0 PSU(n)$. One may be tempted to associate $\nu^{SU(n)}$ to $\nu[\,]$, but this is easily seen to be inconsistent with the tensor product at generic $n$, as we have for instance already seen that the adjoint of $SU(n)$ $[21^{n-2}]$ should rather be associated with the representation $[1]\overline{[1]}$.

# 3 Spin chains and loop models with $PSU(n)$ symmetry

## 3.1 The $U(n)$ and $O(n)$ spin chains

A quantum spin chain of length $L$ with a global symmetry group $G$ is defined by

- a spectrum $\mathcal{S}$, which is a tensor product of representations of $G$:

$$\mathcal{S} = R_1 \otimes \cdots \otimes R_L, \tag{49}$$

- a Hamiltonian $H : \mathcal{S} \to \mathcal{S}$, which commutes with the action of $G$.

**Spectra**

We first assume that the representation $R_i$ is the same at each site, and is built from the identity representation $[\,]$, together with the simplest non-identity representation(s) of $G$:

$$\mathcal{S}^{O(n)} = ([\,] + [1])^{\otimes L}, \tag{50a}$$

$$\mathcal{S}^{U(n)} = ([\,] + [1] + \overline{[1]})^{\otimes L}. \tag{50b}$$

In the spectra $\mathcal{S}^{O(n)}$ and $\mathcal{S}^{U(n)}$, a site can either be empty $[\,]$ or carry a spin $[1]$, or a complex-conjugated spin $\overline{[1]}$ for the $U(n)$ chain.

**Hamiltonians**

We must describe how spins in the chains (50) interact with one another. We assume nearest-neighbour interactions, with periodic boundary conditions. This means that the hamiltonians can be written as

$$H = \sum_{i=1}^{L} H_{i,i+1}, \tag{51}$$

where $H_{i,i+1}$ operates on the $i^{\text{th}}$ and $(i+1)^{\text{th}}$ sites, and the $(L+1)^{\text{th}}$ site is identified with the first. The hamiltonians $H_{i,i+1}$ are $G$-equivariant morphisms

$$([\,] + [1])^{\otimes 2} \to ([\,] + [1])^{\otimes 2}, \quad G = O(n), \tag{52}$$

or

$$([\,] + [1] + \overline{[1]})^{\otimes 2} \to ([\,] + [1] + \overline{[1]})^{\otimes 2}, \quad G = U(n). \tag{53}$$

Such morphisms can be represented as diagrams like those of the Brauer or walled Brauer algebras, with the difference that some of the sites can be empty. A crucial assumption is that the diagrams are planar, i.e. we forbid line crossings. This makes the resulting spin chain simpler, and in fact exactly solvable. Due to the periodic boundary conditions, the following diagrams are considered planar:

$$
\begin{aligned}
e_L &= \quad\cdots\quad \\
u &= \quad\cdots\quad
\end{aligned}
\tag{54}
$$

In the case of the $O(n)$ chain, the planar nearest-neighbour interactions are described by the following 9 diagrams

$$\tag{55}$$

In the case of the $U(n)$ spin chain, we can a priori assign arbitrary orientations to all the lines in these 9 diagrams. For simplicity, we restrict ourselves to an alternating spin chain, i.e. we forbid the following four diagrams:

$$\tag{56}$$

We recover the model of [4], whose Hamiltonians $H_{i,i+1}$ are built from the following 17 diagrams, acting on states with alternating orientations:

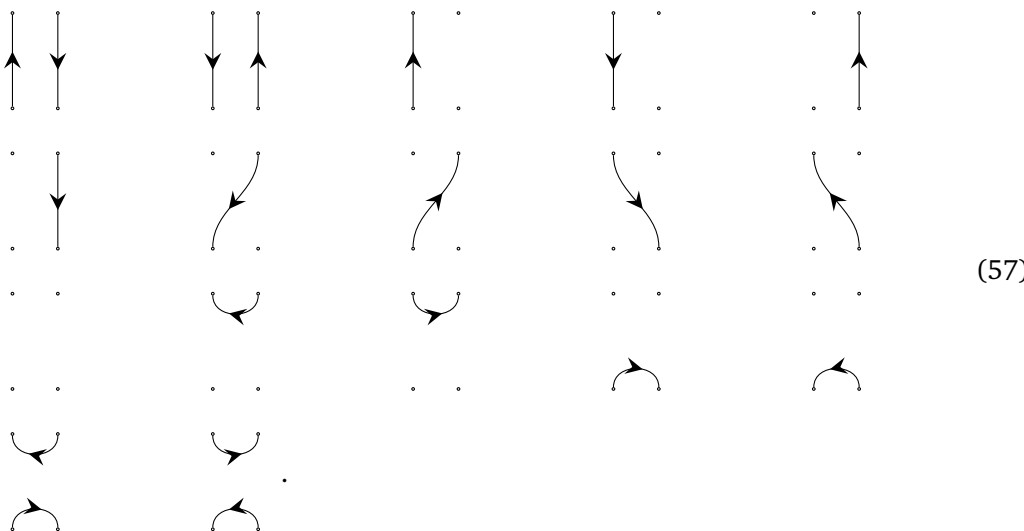

$$\tag{57}$$

**Completely packed case**

We call Completely Packed (CPacked) the case where configurations with empty sites have zero weight. While this case is less general than the dilute versions of the spin chains, it is technically convenient to study, and the critical behaviour is equivalent. In this case, the spectra (50a) and (50b) reduce to

$$S^{O(n),\text{CPacked}} = [1]^{\otimes L}, \tag{58a}$$

$$S^{U(n),\text{CPacked}} = \left([1] \otimes \overline{[1]}\right)^{\otimes \frac{L}{2}}. \tag{58b}$$

The possible Hamiltonians are written in terms of a parameter $x$ called the weight,

$$H_{i,i+1}^{O(n)} = \left| \; \right| \;\; + x \;\; \smile\!\!\frown \tag{59}$$

$$H_{2i-1,2i}^{U(n)} = \uparrow \; \downarrow \;\; + x \qquad\qquad H_{2i,2i+1}^{U(n)} = \downarrow \; \uparrow \;\; + x \tag{60}$$

As explained in the introduction, the hamiltonian on the completely packed chain has $PSU(n)$ symmetry, since $U(1)$ acts trivially.

**Alternative dilution**

We could alternatively consider the spectrum

$$\mathcal{S}^{U(n),\text{ alternative}} = \left(([\;] + [1]) \otimes ([\;] + \overline{[1]})\right)^{\otimes \frac{L}{2}}, \tag{61}$$

and build local hamiltonians $H_{i,i+1}$ from the diagrams in (57), with a weight $x$ per monomer and with suitable orientations depending on the parity of $i$. This model was considered in [26] and its critical point was found to have a larger symmetry group $O(2n) \supset U(n)$.

## 3.2 Lattice loop models

Evolving a one-dimensional spin chain with a discrete time step yields a two-dimensional loop model. Loop models are statistical models whose states are configurations of loops on a lattice. Each loop has a weight that can depend on its topology, and each occupied lattice edge also comes with a weight $x$. We distinguish the following cases:

- A completely packed spin chain leads to a loop model where every edge is covered. The loop model is also called completely packed.

- We call Fully Packed a loop model where every vertex is visited.

- When a model is neither Completely Packed nor Fully Packed, we say it is dilute. This notion of a *dilute model* should not be confused with the *dilute phase* of the $O(n)$ loop model: a phase of a dilute model that also has a dense phase.

**The $O(n)$ loop model**

The loop model corresponding to the spin chain described by the spectrum (50a) and the interactions (55) is the $O(n)$ loop model. While the evolution of the spin chain naturally leads

to a square lattice as pictured in (64), it is convenient to reformulate the loop model on a honeycomb lattice by adding horizontal edges [27]:

$$(62)$$

In the loop model, each loop has a weight $n$, and each covered edge of the hexagonal lattice has a weight $x$. Integrability can then be used for determining critical lines $x(n)$ [28], leading to the model's phase diagram for $x > 0, n \in [-2, 2]$, reproduced in Figure (4).

**The $PSU(n)$ loop model**

The $PSU(n)$ loop model corresponds to the chain described by the spectrum (50b) and the interactions (57) [4].

We say that a configuration of loops is orientable if the loops can be oriented such that two adjacent loops always carry opposite orientations. Equivalently, this means that the lattice can be colored in black and white such that each loop is a boundary between regions of different colors, and turns clockwise around black regions. On a planar graph, any loop configuration is orientable. On the torus, a loop configuration is orientable if and only if each fundamental cycle is crossed an even number of times by non-contractible loops. For instance, the following loop configuration is not orientable if the graph has periodic boundary conditions in both directions:

$$(63)$$

The $PSU(n)$ loop model's states are orientable loop configurations, with the same weights as in the $O(n)$ loop model.

**Completely packed loops on the square lattice**

Evolving the completely packed $U(n)$ spin chain, we obtain a loop model on a square lattice:

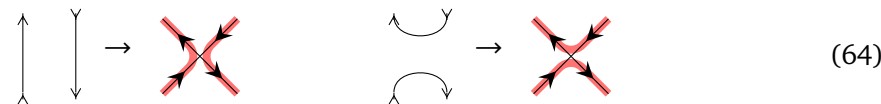

$$(64)$$

Since loops are oriented and cover all lattice edges, the lattice itself is oriented. The resulting lattice orientation is called the Chalker-Coddington orientation:

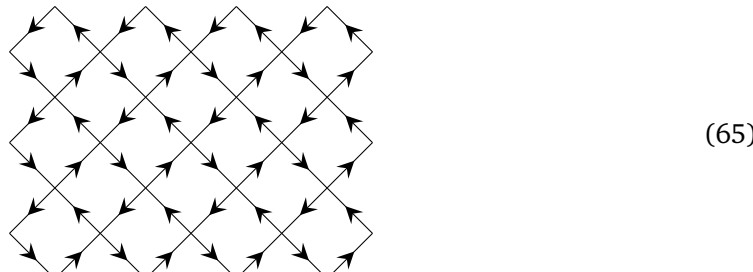

$$(65)$$

An example of a loop configuration on this lattice is found in Figure (82).

**Introduction of crossings**

Few results are known for models with non-planar interactions, let alone exact results. A heuristic argument to predict whether or not introducing crossings changes the universality class is to study the relevance of the operator that inserts a crossing, namely the 4-leg operator $V_{(2,0)}$. This operator is relevant on the dense critical line of the $O(n)$ loop model, and irrelevant on the dilute critical line. Therefore, the dilute phase is expected to be robust against the introduction of crossings, while the dense phase is expected to be unstable [10]. For the $U(n)$ spin chain, non-planar interactions must involve a next-to-nearest neighbour interaction, as studied in [29].

## 4 Spectrum of the $PSU(n)$ CFT

In the limit of large $L$, the spectrum of the $U(n)$ spin chain's hamiltonian (60), or equivalently observables of the $U(n)$ loop model, are described in terms of fields of the $PSU(n)$ CFT. The set of primary fields of the $PSU(n)$ CFT was first determined in [11]. The main technical result of this article is the determination of the action of $PSU(n)$ on these fields. In Sections 4.1 and 4.2, we give two different derivations of the following result:

$$\mathfrak{S}^{PSU(n)} \underset{\mathfrak{C} \times U(n)}{=} \sum_{s \in \mathbb{N}^*} (\mathscr{R}_{\langle 1,s \rangle} \otimes [\,]) + \sum_{(r,s) \in \mathbb{N}^* \times \frac{1}{r}\mathbb{Z}} (\mathscr{W}_{(r,s)} \otimes \Omega_{(r,s)}). \tag{66}$$

Here $\mathscr{R}_{\langle 1,s \rangle}$ and $\mathscr{W}_{(r,s)}$ are representations of the conformal algebra $\mathfrak{C} = \mathrm{Vir} \oplus \overline{\mathrm{Vir}}$ defined in (7), $[\,]$ is the identity representation of $U(n)$, and $\Omega_{(r,s)}$ is a representation of $U(n)$. Throughout this section we use the isomorphism $\underline{\mathrm{Rep}}_0 PSU(n) \simeq \underline{\mathrm{Rep}}_0 U(n)$ described in section 2 to write all representations as representations of $U(n)$ instead of $PSU(n)$.

Our two methods give two very different expressions for the $\Omega_{(r,s)}$:

$$\Omega_{(r,s)} = \delta_{r,1}[\,] + \frac{1}{r} \sum_{r'=0}^{r-1} e^{2i\pi r's} \tilde{T}_{r \wedge r'} \left( \omega_{\frac{r}{r \wedge r'}} \otimes \overline{\omega}_{\frac{r}{r \wedge r'}} - 2 \right) \tag{67}$$

$$= \sum_{\lambda\bar{\mu}} c^{\lambda\bar{\mu}}_{(r,s)} \lambda\bar{\mu}, \tag{68}$$

where

- $\tilde{T}_d$ is the Chebyshev polynomial such that

$$\tilde{T}_d(X + X^{-1}) = X^d + X^{-d}, \tag{69}$$

- $\omega_m$ is the formal representation of $U(n)$ whose character is the power-$m$ polynomial $z_1^m + \cdots + z_n^m$. For instance, $\omega_1 = [1]$.

- $r \wedge r'$ is the greatest common divisor of $r$ and $r'$, with the convention $r \wedge 0 = r$.

- $c_{(r,s)}^{\lambda\bar\mu} \in \mathbb{N}$ is a branching coefficient for which we give a combinatorial formula in (112).

The agreement of our two expressions is far from manifest, and can only be tested in examples. The representations $\Omega_{(r,s)}$ obey the relations

$$\Omega_{(r,s+1)} = \Omega_{(r,s)}, \tag{70a}$$
$$\Omega_{(r,-s)} = \Omega_{(r,s)}, \tag{70b}$$

which allow us to restrict our attention to values $s \in \frac{1}{r}\mathbb{Z} \cap [0, \frac{1}{2}]$ of the second Kac index. Let us display the first 6 examples by increasing values of $r$ then $s$:

$$\Omega_{(1,0)} = [1]\overline{[1]}, \tag{71a}$$

$$\Omega_{(2,0)} = [2]\overline{[2]} + [1^2]\overline{[1^2]}, \tag{71b}$$

$$\Omega_{(2,\frac{1}{2})} = [2]\overline{[1^2]} + [1^2]\overline{[2]}, \tag{71c}$$

$$\Omega_{(3,0)} = [3]\overline{[3]} + [3]\overline{[1^3]} + [1^3]\overline{[3]} + 2[21]\overline{[21]} + [2]\overline{[2]} + [2]\overline{[1^2]} + [1^2]\overline{[2]} + [1^3]\overline{[1^3]}$$
$$+ [1^2]\overline{[1^2]} + [1]\overline{[1]} + [\,]\overline{[\,]}, \tag{71d}$$

$$\Omega_{(3,\frac{1}{3})} = [3]\overline{[21]} + [21]\overline{[3]} + [21]\overline{[21]} + [2]\overline{[2]} + [21]\overline{[1^3]} + [1^3]\overline{[21]} + [2]\overline{[1^2]} + [1^2]\overline{[2]}$$
$$+ [1^2]\overline{[1^2]} + [1]\overline{[1]}, \tag{71e}$$

$$\Omega_{(4,0)} = [4]\overline{[4]} + [4]\overline{[2^2]} + [2^2]\overline{[4]} + [4]\overline{[21^2]} + [21^2]\overline{[4]} + 3[31]\overline{[31]} + 2[3]\overline{[3]}$$
$$+ [31]\overline{[2^2]} + [2^2]\overline{[31]} + 2[31]\overline{[21^2]} + 2[21^2]\overline{[31]} + [31]\overline{[1^4]} + [1^4]\overline{[31]}$$
$$+ 2[2^2]\overline{[2^2]} + [2^2]\overline{[21^2]} + [21^2]\overline{[2^2]} + 3[21^2]\overline{[21^2]} + [2^2]\overline{[1^4]} + [1^4]\overline{[2^2]} + [1^4]\overline{[1^4]}$$
$$+ 2[3]\overline{[1^3]} + 2[1^3]\overline{[3]} + 8[21]\overline{[21]} + 4[21]\overline{[1^3]} + 4[1^3]\overline{[21]} + 2[1^3]\overline{[1^3]}$$
$$+ 6[2]\overline{[2]} + 4[2]\overline{[1^2]} + 4[1^2]\overline{[2]} + 6[1^2]\overline{[1^2]} + 4[1]\overline{[1]} + 2[\,]\overline{[\,]}. \tag{71f}$$

## 4.1 Twisted torus partition function

**Definition**

The partition function of a conformal field theory on a torus of modulus $\tau$ is defined from the spectrum $\mathcal{S}$ by

$$Z = \mathrm{Tr}_{\mathcal{S}}\left(e^{2i\pi\tau(L_0 - \frac{c}{24})} e^{-2i\pi\bar\tau(\bar L_0 - \frac{c}{24})}\right). \tag{72}$$

The spectrum is decomposed into representations $\mathcal{W} \otimes R$ of $\mathfrak{C}_c \times U(n)$ as in Eq. (66), where $\mathfrak{C}_c$ is the conformal algebra with central charge $c$, i.e. two copies of the Virasoro algebra Vir, called left-moving and right-moving:

$$\mathfrak{C}_c = \mathrm{Vir} \oplus \overline{\mathrm{Vir}}. \tag{73}$$

Consequently $Z$ is a sum of terms of the type $\chi_{\mathcal{W}}(\tau)\dim R$, where $\chi_{\mathcal{W}}$ is the character of $\mathcal{W}$:

$$\chi_{\mathcal{W}} = \mathrm{Tr}_{\mathcal{W}}\left(e^{2i\pi\tau(L_0 - \frac{c}{24})} e^{-2i\pi\bar\tau(\bar L_0 - \frac{c}{24})}\right). \tag{74}$$

This character tells us which primary states appear in $\mathscr{W}$, and this is enough for identifying $\mathscr{W}$ among the representations that we considered in (7). Note that the logarithmic nature of the representations $\mathscr{W}_{(r,s)}$ when $r, s \in \mathbb{N}^*$ was known beforehand, and cannot be deduced from the torus partition function. Indeed, as is shown in [15] the representation $\mathscr{W}_{(r,s)}$ has the same character as a direct sum of Verma modules:

$$\chi_{\mathscr{W}_{(r,s)}} = \chi_{\mathscr{V}_{(r,s)}} + \chi_{\mathscr{V}_{(r,-s)}}. \tag{75}$$

On the other hand, the partition function only tells us about the dimension of $U(n)$ representations, which is in general not enough for determining them. We remedy this by considering the twisted partition function

$$Z(g) = \mathrm{Tr}_{\mathcal{S}}\left(e^{2i\pi\tau(L_0 - \frac{c}{24})}e^{-2i\pi\bar{\tau}(\bar{L}_0 - \frac{c}{24})}g\right), \tag{76}$$

where $g \in U(n)$. This corresponds to inserting an element $g \in U(n)$ at a constant-time slice on the torus, which we now draw as a parallelogram $\frac{\mathbb{C}}{\mathbb{Z}+\tau\mathbb{Z}}$ with opposite sides identified:

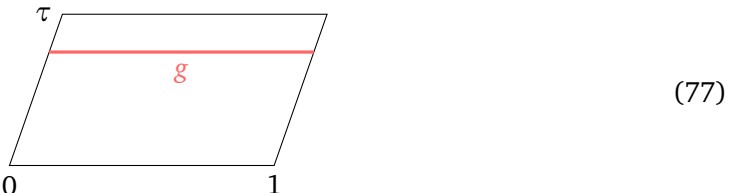

$$\tag{77}$$

The line of insertion of $g$ is topological, i.e. it can be deformed without changing $Z(g)$. Group elements $g \in U(n)$ in principle only make sense for $n \in \mathbb{N}$, but in the end our expressions will involve characters of representations of $U(n)$, which can be interpreted for complex $n$ in the universal character ring of $U(n)$. A representation $\mathscr{W} \otimes R \subset \mathcal{S}$ of $\mathfrak{C}_c \times U(n)$ now contributes a term $\chi_{\mathscr{W}}(\tau)\chi_R(g)$ to the twisted partition function, and from its character $\chi_R(g)$ we can deduce the representation $R$ of $U(n)$. Note that the insertion of the twist breaks modular invariance, since the modular transform of $Z(g)$ corresponds to the partition function with an insertion of $g$ along a vertical cycle of the torus, which cannot be obtained by topological deformations of the insertion on horizontal cycle. This is not an issue since our method for computing $Z$ does not rely on modular invariance.

A simple example of a twist by a global symmetry is found in the Ising model. The Ising model has a global $\mathbb{Z}_2$ symmetry $\sigma \to -\sigma$. The corresponding $\mathbb{Z}_2$-twist is implemented by the defect operator of [30]. This $\mathbb{Z}_2$-twist encodes the action of the global $\mathbb{Z}_2$ symmetry on the spectrum of the corresponding minimal model: $\{1, \sigma, \epsilon\} \mapsto \{1, -\sigma, \epsilon\}$.

Next we will compute $Z(g)$ by expressing it as a linear combination of partition functions for free bosons, which are explicitly known in the critical limit [11]. To make contact with free bosons, we will however have to rewrite our unoriented (though orientable) loop model in terms of oriented loops.

**From unoriented to oriented loop models**

Let us assume that each loop can have 2 orientations, with fugacities of the type $\mathfrak{q}$ and $\mathfrak{q}^{-1}$. This amounts to rewriting the loop weight as

$$n = -2\cos\pi\beta^2 = \mathfrak{q} + \mathfrak{q}^{-1}, \quad \text{with} \quad \mathfrak{q} = -e^{i\pi\beta^2}. \tag{78}$$

Then the partition function of a loop model on a certain ensemble $\mathscr{E}_u$ of unoriented loops

$$Z = \sum_{\mathscr{L} \in \mathscr{E}_u} n^{|\mathscr{L}|}, \tag{79}$$

is rewritten as a sum over the corresponding ensemble $\mathscr{E}$ of oriented loops as

$$Z = \sum_{\mathscr{L} \in \mathscr{E}} \mathfrak{q}^{|\mathscr{L}_+| - |\mathscr{L}_-|} = \frac{1}{2} \sum_{\mathscr{L} \in \mathscr{E}} \tilde{T}_{|\mathscr{L}_+| - |\mathscr{L}_-|}(n). \tag{80}$$

To be clear, any unoriented configuration in $\mathscr{E}_u$ with $k$ loops gives rise to $2^k$ oriented loop configurations $\mathscr{L} \in \mathscr{E}$, with $|\mathscr{L}_\pm|$ the number of loops of each orientation. The second equality of (80) makes use of the symmetry $\mathfrak{q} \to \mathfrak{q}^{-1}$ to rewrite the sum in terms of Chebyshev polynomials.

Beware that the orientations that we just introduced have nothing to do with the *global* orientation of loops in the completely packed loop model, which is an orientation of the underlying graph and is fixed, not summed over.

**Twisted partition function of the completely packed loop model**

We consider a completely packed loop model on a torus, with a row where a twist by $g \in U(n)$ is performed, as in Figure (77). The twist is well-defined in the spin chain, where $g$ acts on spins by simultaneously rotating them all. In the loop model, this corresponds to modifying the fugacities of some of the loops. Let us describe what happens to various types of loops. (See Figure (82) for an example.)

- Loops that do not cross the twisted row are unaffected, keeping weight $n$.

- A contractible loop that intersects the twisted row necessarily contains as many twisted $\uparrow$ edges as $\downarrow$ edges. Thus, the weight of the loop picks as many $g$ as $g^\dagger$ factors, which cancel out after integrating over the spins, meaning the weight of the loop remains $n$.

- For the same reason, loops that only wind horizontally around the torus, i.e. around the $(0, 1) \in \pi_1(\mathbb{T})$ cycle, keep their weight $n$. By an abuse of language, we also call such loops contractible.

- There remain those loops that wind around the $(1, 0)$ cycle of the torus, drawn in blue in figure (82). Let us introduce the numbers of times the loop winds around the $(1, 0)$ and $(0, 1)$ cycles:

$$(m, m') \in \mathbb{N}^2. \tag{81}$$

Because the loop is non-self-intersecting, $m \wedge m' = 1$ where $\wedge$ denotes the greatest common divisor. Two scenarios can occur: either the loop crosses $m$ more $\uparrow$ than $\downarrow$ twisted edges, in which case its weight becomes $\mathrm{Tr}_{[1]} g^m$ and we say the loop is of type $\uparrow$, or it crosses $m$ less $\uparrow$ than $\downarrow$ twisted edges, in which case its weight becomes $\mathrm{Tr}_{[1]}(g^\dagger)^m = \mathrm{Tr}_{\overline{[1]}} g^m$ and we say the loop is of type $\downarrow$. Since the model is completely packed there has to be as many $\uparrow$ as $\downarrow$ loops in all loop configurations. Also, because loops are non-intersecting, all non-contractible loops in a given configuration $\mathscr{L}$ have the same $(m(\mathscr{L}), m'(\mathscr{L}))$.

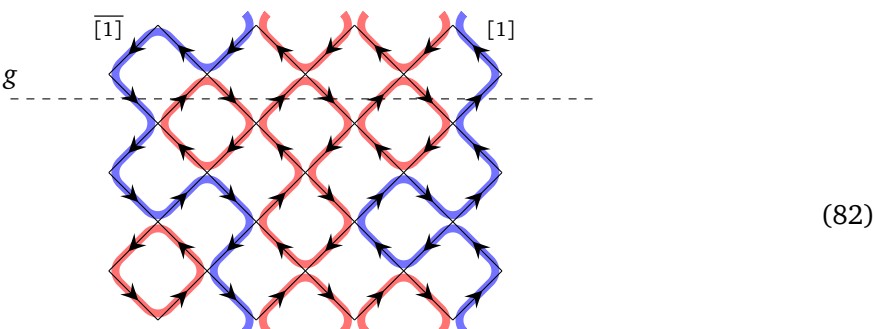

$$(82)$$

A configuration of completely packed loops with $m = 1$, $m' = 0$, $N_0 = 6$, $N = 2$.

Let $N_0(\mathscr{L})$ (resp. $N(\mathscr{L})$) be the number of contractible (resp. non-contractible) loops in a given loop configuration $\mathscr{L} \in \mathscr{E}_u$, we see that the twisted partition function for the completely packed loop model is

$$
\begin{aligned}
Z(g) &= \sum_{\mathscr{L} \in \mathscr{E}_u} n^{N_0(\mathscr{L})} \left( \mathrm{Tr}_{[1]} g^{m(\mathscr{L})} \right)^{N(\mathscr{L})/2} \left( \mathrm{Tr}_{\overline{[1]}} g^{m(\mathscr{L})} \right)^{N(\mathscr{L})/2} \\
&= \sum_{\mathscr{L} \in \mathscr{E}_u} n^{N_0(\mathscr{L})} \sqrt{\mathrm{Tr}_{[1] \otimes \overline{[1]}} g^{m(\mathscr{L})}}^{N(\mathscr{L})},
\end{aligned}
$$
$$(83)$$

where $\mathscr{E}_u$ is an ensemble of unoriented loops.

**Relation to frustrated free bosons and critical limit**

Let us rewrite our twisted partition function $Z(g)$ as a sum over oriented loops. Let $N_{\pm}^0$ ($N_{\pm}$) be the number of (non) contractible loops of each orientation, and let

$$
M = m(N_+ - N_-), \quad M' = m'(N_+ - N_-),
$$
$$(84)$$

be the total winding numbers around each cycle of the torus. Because the number $N$ of non-contractible loops is always even in the completely packed loop model, so are $M$ and $M'$. Since $m \wedge m' = 1$,

$$
M \wedge M' = |N_+ - N_-|,
$$
$$(85)$$

and $m = \frac{M}{M \wedge M'}$. By orienting the loops and using (80), we find

$$
Z(g) = \frac{1}{2} \sum_{M, M' \in 2\mathbb{Z}} \underbrace{\left( \sum_{\substack{\mathscr{L} \in \mathscr{E} \\ M(\mathscr{L}) = M, M'(\mathscr{L}) = M'}} q^{N_+^0 - N_-^0} \right)}_{=: Z_{M,M'}} \tilde{T}_{M \wedge M'} \left( \sqrt{\mathrm{Tr}_{[1] \otimes \overline{[1]}} g^{\frac{M}{M \wedge M'}}} \right).
$$
$$(86)$$

Seeing the oriented loops as level lines for a height field, we identify the term $Z_{M,M'}$ as the partition function of a height field $h$ with boundary conditions $h(x+1, t) = h(x, t) + M h_0$ and $h(x, t + \tau) = h(x, t) + M' h_0$. This field is known to renormalise to a compact free boson in the critical limit,

$$
Z_{M,M'} \xrightarrow[\text{critical}]{} \frac{\beta}{2\sqrt{\Im \tau} |\eta(\tau)|^2} e^{-\frac{\pi \beta^2}{4 \Im \tau}(M^2 |\tau|^2 - 2MM' \Re \tau + M'^2)}.
$$
$$(87)$$

Now let us consider the $M = 0$ term in the critical limit of the twisted partition function,

$$Z^{\text{critical}}(g)\big|_{M=0} = \frac{1}{2} \sum_{M' \in 2\mathbb{Z}} \frac{\beta}{2\sqrt{\Im\tau}|\eta(\tau)|^2} e^{-\frac{\pi\beta^2}{4\Im\tau}M'^2} \tilde{T}_{M'}(n). \tag{88}$$

We use $\tilde{T}_{M'}(n) = e^{i\pi M'(\beta^2+1)} + e^{-i\pi M'(\beta^2+1)} = 2\cos(\pi M'(\beta^2 + 1))$, and perform a Poisson resummation, which replaces $M'$ with a new variable $s$. We obtain a combination of characters of the representations from (7):

$$Z^{\text{critical}}(g)\big|_{M=0} = \frac{1}{2}\left\{ \sum_{s \in \mathbb{N}^*} \chi^d_{\langle 1,s \rangle}(\tau) + \sum_{s \in \mathbb{Z}} \chi_{(1,s)}(\tau) \right\}. \tag{89}$$

For the $M \neq 0$ terms, we write $M = 2r$, $M' = kM + 2r'$ where $r' = 0, \ldots, r - 1$, such that $M \wedge M' = 2(r \wedge r')$. After Poisson transforming the sum over $k$, using $\tilde{T}_{2d} = \tilde{T}_d(X^2 - 2)$ and some relabelling, we find

$$Z^{\text{critical}}(g)\big|_{M \neq 0} = \frac{1}{2}\left\{ \sum_{r \geq 1} \sum_{s \in \frac{1}{r}\mathbb{Z}} \chi_{(r,s)}(\tau) \frac{1}{r} \sum_{r'=0}^{r-1} e^{2i\pi sr'} \tilde{T}_{r \wedge r'}\left( \text{Tr}_{[1]\otimes\overline{[1]}} g^{\frac{r}{r \wedge r'}} - 2 \right) \right\}. \tag{90}$$

The complete partition function is the sum of the $M = 0$ and $M \neq 0$ terms.

**Action of $U(n)$ on the spectrum**

In our twisted partition function, it remains to identify the coefficient of a conformal character $\chi_{(r,s)}(\tau)$ as the characters of some $U(n)$ representation $\Omega_{(r,s)}$:

$$\chi_{\Omega_{(r,s)}}(g) = \delta_{r,1} + \frac{1}{r} \sum_{r'=0}^{r-1} e^{2i\pi sr'} \tilde{T}_{r \wedge r'}\left( \text{Tr}_{[1]\otimes\overline{[1]}} g^{\frac{r}{r \wedge r'}} - 2 \right), \tag{91}$$

where $\delta_{r,1}$ comes from (89) and the rest from (90). We introduce a formal representation

$$\omega_m = \sum_{k=0}^{m-1} (-1)^k [m-k, 1^k][\overline{\phantom{]}}]. \tag{92}$$

As is well-known, the character of $\omega_m$ is the inductive limit of the power-$m$ polynomial: for $g \in U(n)$, $\chi_{\omega_m}(g) = \text{Tr}_{[1]} g^m$. This can be seen as a special case of the Murnaghan–Nakayama rule, see [31](Section 8) or [32](Theorem 21.4). It is then immediate that (91) is the character of $\Omega_{(r,s)}$ as defined in (67).

At this point it is not manifest that $\Omega_{(r,s)}$ is a representation of $U(n)$, i.e. a linear combination of irreducible representations with coefficients in $\mathbb{N}$. Rather, it is written as a formal representation, whose coefficients are a priori in $\mathbb{C}$. Using Ramanujan sums $\varphi_k(r) = \sum_{l=1}^{r} \delta_{r \wedge l,1} e^{2i\pi\frac{kl}{r}} \in \mathbb{Z}$, we rewrite

$$\Omega_{(r,s)} = \delta_{r,1}[\,] + \frac{1}{r} \sum_{\substack{a;b|r \\ ab=r}} \varphi_{rs}(a) \tilde{T}_b(\omega_a \otimes \bar{\omega}_a - 2), \tag{93}$$

which now has coefficients in $\frac{1}{r}\mathbb{Z}$. We have no proof that these coefficients are actually in $\mathbb{N}$, although we have checked it explicitly for $r \leq 6$ using our computer code [18].

## 4.2 Branching rules of diagram algebras

Let us study the action of symmetries on the spectrum $\mathcal{S}_L^{U(n)}$ (58b) of a spin chain of finite length $L$. While the conformal algebra only appears in the limit $L \to \infty$, it has a finite $L$ counterpart called the Jones–Temperley–Lieb algebra $\mathscr{JTL}_L(n)$. Decomposing the spectrum into representations of $\mathscr{JTL}_L(n) \times U(n)$, we will find that the representations $\Omega_{(r,s)}$ already emerge at finite $L$, provided $L \geq r$.

**Decomposition of the space of states under $\mathscr{B}_{L,L}(n)$**

We have introduced the walled Brauer algebra $\mathscr{B}_{L,L}(n)$ as the set of endomorphisms of $\mathcal{S}_L^{U(n)}$ that commute with $U(n)$. Conversely, endomorphisms of $\mathcal{S}_L^{U(n)}$ that commute with $\mathscr{B}_{L,L}(n)$ belong to $U(n)$. According to Schur–Weyl duality [33], it follows that the space of states $\mathcal{S}_L^{U(n)}$ completely factorises as a bi-module under the joint action of $U(n)$ and $\mathscr{B}_{L,L}(n)$:

$$\mathcal{S}_L^{U(n)} \underset{\mathscr{B}_{L,L}(n) \times U(n)}{=} \sum_{\substack{\lambda,\mu \\ |\lambda|,|\mu| \leq L \\ |\lambda|=|\mu|}} B_{L,L}^{\lambda\bar{\mu}} \otimes \lambda\bar{\mu} \,. \tag{94}$$

However, the walled Brauer algebra is not physical, in the sense that it is not planar: it allows sites to interact irrespective of their distance, and therefore does not know about the spatial structure of the spin chain. The physical algebra is the Jones-Temperley-Lieb subalgebra $\mathscr{JTL}_L(n) \subset \mathscr{B}_{L,L}(n)$. We should therefore further decompose $B_{L,L}^{\lambda\bar{\mu}}$ into representations of $\mathscr{JTL}_L(n)$.

**Representations of the Jones-Temperley-Lieb algebra $\mathscr{JTL}_L(n)$**

The Jones-Temperley-Lieb algebra $\mathscr{JTL}_L(n)$ has generators $e_i, i \in \mathbb{Z}_{2L}$ and $u^2$, with relations

$$e_i e_{i\pm1} e_i = e_i \,, \tag{95a}$$

$$u^2 e_i u^{-2} = e_{i+2} \,, \tag{95b}$$

$$u^2 e_{2L-1} = e_1 e_2 \cdots e_{2L-1} \,, \tag{95c}$$

$$u^{2L} = 1 \,. \tag{95d}$$

Elements of $\mathscr{JTL}_L(n)$ can be drawn as planar diagrams, i.e. as diagrams where lines do not intersect, provided we allow for lines to cross the periodic boundary condition as in (54). For instance, $u^2$ is drawn as



$$\tag{96}$$

Like for the walled Brauer algebra, modules of the $\mathscr{JTL}_L(n)$ algebra are described in terms of planar link patterns with $2r$ defect lines, $r \leq L$, i.e. link patterns whose links do no intersect. Instead of carrying a representation of the symmetric groups $S_{|\lambda|} \times S_{|\mu|}$ as in the walled Brauer case, the link patterns now carry a representation of the cyclic group $\mathbb{Z}_r$. This is because of planarity and orientation constraints, which mean that the action of $\mathscr{JTL}_L(n)$ can only cyclically permute the $r$ pairs of $\uparrow\downarrow$ defect lines. For $s \in (\frac{1}{r}\mathbb{Z})/\mathbb{Z}$, let $Z_s^r$ be the (one-dimensional) representation of $\mathbb{Z}_r$ where the generator of $\mathbb{Z}_r$ acts as $e^{2i\pi s}$, then the representation of $\mathscr{JTL}_L(n)$ with $r \in \mathbb{N}$ pairs of defect lines and the defect momentum $s \in (\frac{1}{r}\mathbb{Z})/\mathbb{Z}$ is

$$W_{(r,s)}^L = \mathbb{C}p_r^L \otimes_{\mathbb{Z}_r} Z_s^r \,, \tag{97}$$

where we define

$$p_r^L = \left\{\text{planar link patterns with } 2L \text{ alternating sites and } r \text{ pairs of } \uparrow\downarrow \text{ defect lines}\right\}. \quad (98)$$

We call $t^2$ the *pseudo-translation* by two sites, which is the operator that cyclically permutes the pairs of $\uparrow\downarrow$ defect lines in $W_{(r,s)}^L$. Notice that $t^2 \in \mathbb{Z}_r$ should not be confused with $u^2 \in \mathbb{Z}_L$ which translates the actual sites by two units. By definition, $t^2$ acts as $e^{2i\pi s}$ in $W_{(r,s)}^L$:

$$t^2 W_{(r,s)}^L = e^{2i\pi s} W_{(r,s)}^L. \quad (99)$$

**Decomposition of the space of states under $\mathscr{JTL}_L(n)$**

Because $\mathscr{JTL}_L(n)$ is a subalgebra of $\mathscr{B}_{L,L}(n)$, modules of the walled Brauer algebra decompose into modules of the $\mathscr{JTL}_L(n)$ algebra. Suppose we know the branching coefficients $c_{(r,s)}^{\lambda\bar{\mu}} \in \mathbb{N}$ such that

$$B_{L,L}^{\lambda\bar{\mu}} \underset{\mathscr{JTL}_L(n)}{=} \sum_{r,s} c_{(r,s)}^{\lambda\bar{\mu}} W_{(r,s)}^L, \quad (100)$$

and that these coefficients are independent of $L$. Then (94) gives

$$\mathcal{S}_L^{U(n)} \underset{\mathscr{JTL}_L(n) \times U(n)}{=} \sum_{(r,s)} W_{(r,s)}^L \otimes \underbrace{\left(\sum c_{(r,s)}^{\lambda\bar{\mu}} \lambda\bar{\mu}\right)}_{\Omega_{(r,s)}}. \quad (101)$$

To get the spectrum of the $PSU(n)$ CFT, it remains to take the critical limit. From lattice computations [34] we know that the $\mathscr{JTL}_L(n)$ algebra is a discretisation of the so called interchiral algebra $\widetilde{\mathfrak{C}}_{\beta^2}$, which is the algebra generated by the modes of the stress-energy tensors $T(z), \bar{T}(\bar{z})$ and of the degenerate field $V_{\langle 1,2\rangle}^d$ [34]. Therefore, modules of the $\mathscr{JTL}_L(n)$ algebra become modules of the interchiral algebra in the critical limit:

$$\lim_{L\to\infty} W_{(r,s)}^L = \widetilde{\widetilde{\mathscr{W}}}_{(r,s)}, \quad (102)$$

where the module $\widetilde{\widetilde{\mathscr{W}}}_{(r,s)}$ of the interchiral algebra is an infinite sum of modules of the Virasoro algebra, as dictated by the fusion rules of $V_{\langle 1,2\rangle}$:

$$\widetilde{\widetilde{\mathscr{W}}}_{(r,s)} = \sum_{n\in\mathbb{Z}} \mathscr{W}_{(r,s+n)}. \quad (103)$$

Consequently, taking the critical limit of the decomposition (101) gives the non-diagonal part of the spectrum of the $PSU(n)$ model (66) and the expression (68) for $\Omega_{(r,s)}$.

**Branching rules $\mathscr{B}_{L,L}(n) \downarrow \mathscr{JTL}_L(n)$**

Let us compute the coefficients $c_{(r,s)}^{\lambda\bar{\mu}}$ of Eq. (100). By analogy with the cases of the Potts and $O(n)$ models [17], we conjecture that these coefficients do not depend on $L$. On the other hand, there is a truncation phenomenon $W_{(r,s)}^{L<r} = 0$, since we cannot have more defects than sites in the chain. We have checked numerically that this $L$-independence holds in numerous examples. To determine $c_{(r,s)}^{\lambda\bar{\mu}}$, it is therefore enough to consider the lowest possible value of $L$ such that $W_{(r,s)}^L$ exists, i.e. $L = r$. The problem then restricts to counting how many copies of $W_{(r,s)}^r$ there are inside $B_{r,r}^{\lambda\bar{\mu}}$. Acting on $W_{(r,s)}^r$, the translation $u^2$ and pseudo-translation $t^2$ coincide on the only relevant link pattern,

$$p_r^r = \left\{ \quad \Big\uparrow \; \Big\downarrow \; \cdots \; \Big\uparrow \; \Big\downarrow \quad \right\}. \quad (104)$$

This implies

$$B_{r,r}^{\lambda\bar{\mu}} = \mathbb{C} d_{|\lambda|,|\mu|}^{r,r} \otimes_{S_r \times S_r} (\lambda \otimes \mu), \tag{105}$$

$$W_{(r,s)}^r \underset{\mathbb{Z}_r}{=} Z_s^r, \tag{106}$$

where the set of link patterns $d_{|\lambda|,|\mu|}^{r,r}$ is defined in Eq. (45).

For example, consider $B_{3,3}^{[2]\overline{[1^2]}}$. It has dimension 9, since the Specht modules $[2]$ and $[1^2]$ have dimension 1. It is spanned by the elements of $d_{2,2}^{3,3}$,

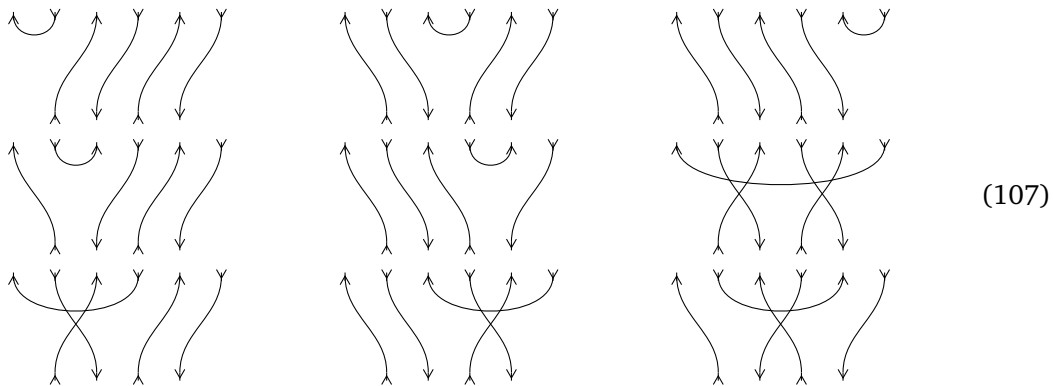

$$(107)$$

Moreover, all $e_i$'s act as zero inside $W_{(r,s)}^r$, since this module is made of link patterns with only defects, which can't be contracted. This gives a characterization of $W_{(r,s)}^r$ as a subrepresentation of $B_{r,r}^{\lambda\bar{\mu}}$; a vector of $B_{r,r}^{\lambda\bar{\mu}}$ is in $W_{(r,s)}^r$ if and only if it is annihilated by all $e_i, i = 1, \ldots, 2L$, and is an eigenvector of $u^2$ of eigenvalue $e^{2i\pi s}$. For any link pattern $d \in d_{|\lambda|,|\mu|}^{r,r}$, if the sites $i$ and $i+1$ are connected then $e_i \cdot d \neq 0$. Hence the link patterns annihilated by all $e_i$'s are those where no two neighbouring sites are connected. Call the set of these link patterns $\ell_{|\lambda|,|\mu|}^{r,r}$.

In our example of $B_{3,3}^{[2]\overline{[1^2]}}$ (107), because the first 6 diagrams have connected neighbours, they are not annihilated by the corresponding $e_i$. So $\ell_{2,2}^{3,3}$ is made of the last three diagrams.

In summary, the number of copies of $W_{(r,s)}^L$ inside $B_{L,L}^{\lambda\bar{\mu}}$ is exactly the dimension of the eigenspace for the eigenvalue $e^{2i\pi s}$ of $u^2$ inside

$$\ell_{|\lambda|,|\mu|}^{r,r} \otimes_{S_r \times S_r} (\lambda \otimes \mu). \tag{108}$$

The action of $u^2$ splits the diagrams in $\ell_{|\lambda|,|\mu|}^{r,r}$ into orbits. Let us focus on a link pattern $d \in \ell_{|\lambda|,|\mu|}^{r,r}$ whose orbit under $u^2$ has length $\omega$. When acting on $d \in B_{L,L}^{\lambda\bar{\mu}}$, since $(u^2)^\omega$ leaves the link pattern invariant, its action reduces to a pair of cyclic permutation $(\sigma, \sigma) \in S_r \times S_r$ of the $\uparrow$ defects between themselves, and the $\downarrow$ defects between themselves. The permutation $\sigma$ has the same order as $(u^2)^\omega$, i.e. $r/\omega$.

For instance, in $B_{4,4}^{[2]\overline{[1^2]}}$ take the link pattern

$$d \quad = \qquad$$ 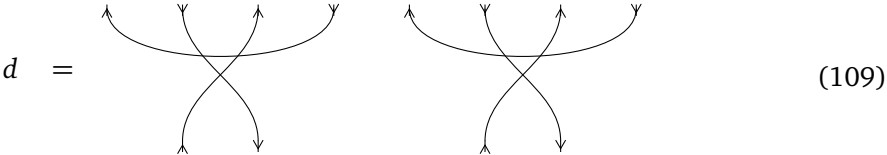 $$\tag{109}$$

Its orbit under $u^2$ is of length $\omega = 2$, and $u^4$ permutes the two $\uparrow$ defects between themselves, and the two $\downarrow$ defects between themselves; $u^4 \cdot d = ((12),(12)) \cdot d$, where $((12),(12)) \in S_2 \times S_2$

acts on the diagram from below, i.e. permutes defect lines. The transposition $(12)$ is indeed of order $2 = r/\omega$.

When focusing on a single orbit, the problem of counting the number of copies of $W^r_{(r,s)}$ inside $B^{\lambda\bar{\mu}}_{r,r}$ thus boils down to counting the multiplicity of the representation $Z^{r/\omega}_s$ of the cyclic group $\mathbb{Z}_{r/\omega} \subseteq S_r \times S_r$ generated by $(\sigma, \sigma)$ inside the product of Specht modules $\lambda \otimes_{S_r \times S_r} \mu$. This boils down to determining branching rules $S_r \downarrow \mathbb{Z}_{r/\omega}$, which fortunately has a known combinatorial solution [35], and yields the following formula for branching rules $S_r \times S_r \downarrow \mathbb{Z}_{r/\omega}$:

$$\lambda \otimes \mu \underset{\mathbb{Z}_{r/\omega}}{=} \sum_{T \in T_\lambda, T' \in T \in T_\mu} Z^r_{\frac{\omega}{r}(\text{ind}\, T + \text{ind}\, T')}, \tag{110}$$

where $\mathbb{Z}_{r/\omega} \hookrightarrow S_r \times S_r$ via $\sigma \mapsto (\sigma, \sigma)$, $T_\lambda$ is the set of tableaux of shape $\lambda$, i.e. the set of all possible numberings of the boxes of the tableau with numbers $1, \ldots, |\lambda|$, such that rows and columns are increasing, and $\text{ind}(T)$ is a combinatorial quantity called the major index of the tableau $T$. The total number of copies of $W^r_{(r,s)}$ in the space generated by the orbit of length $\omega$ is thus

$$\sum_{T \in T_\lambda} \sum_{T' \in T_\mu} \delta_{e^{2i\pi\omega(s - \frac{1}{r}(\text{ind}\, T + \text{ind}\, T'))}, 1}. \tag{111}$$

It just remains to sum over orbits in $\ell^{r,r}_{|\lambda|,|\mu|}$ to get our branching coefficient $c^{\lambda\bar{\mu}}_{(r,s)}$. We call $\mathcal{O}^{|\lambda|}_r = \mathcal{O}^{|\mu|}_r$ the tuple of the lengths of $u^2$-orbits in $\ell^{r,r}_{|\lambda|,|\mu|}$. In our example of $\ell^{3,3}_{2,2}$ the three diagrams are in the same orbit of length 3 under $u^2$, so $\mathcal{O}^2_3 = \{3\}$. We can finally write the branching coefficient

$$c^{\lambda\bar{\mu}}_{(r,s)} = \sum_{\omega \in \mathcal{O}^{|\lambda|}_r} \sum_{T \in T_\lambda} \sum_{T' \in T_\mu} \delta_{e^{2i\pi\omega(s - \frac{1}{r}(\text{ind}\, T + \text{ind}\, T'))}, 1}. \tag{112}$$

Let us give a few examples:

- For $r = 1$, $\mathcal{O}^0_1 = \emptyset$ since the only link state has two neighbouring sites connected, so $c^{[\,]}_{(1,0)} = 0$. $\mathcal{O}^1_1 = \{1\}$ since there is a single link state, so $c^{[1]\overline{[1]}}_{(1,0)} = 1$.

- For $r = 2$, $\mathcal{O}^0_2, \mathcal{O}^1_2 = \emptyset$, so $c^{[\,]}_{(2,s)} = c^{[1]\overline{[1]}}_{(2,s)} = 0$. $\mathcal{O}^2_2 = \{1\}$. There are four possibilities for $\lambda\bar{\mu}$: $\lambda\bar{\mu} \in \{[2]\overline{[2]}, [11]\overline{[11]}, [2]\overline{[11]}, [11]\overline{[2]}\}$. In each case there is a single standard tableau, with $\text{ind}[2] = 0, \text{ind}[11] = 1$. This gives $c^{[2]\overline{[2]}}_{(2,0)} = 1 = c^{[11]\overline{[11]}}_{(2,0)}, c^{[2]\overline{[2]}}_{(2,1/2)} = 0 = c^{[11]\overline{[11]}}_{(2,1/2)},$ $c^{[2]\overline{[11]}}_{(2,1/2)} = c^{[11]\overline{[2]}}_{(2,1/2)} = 1, c^{[2]\overline{[11]}}_{(2,0)} = c^{[11]\overline{[2]}}_{(2,0)} = 0.$

- When $r = 3$, $\mathcal{O}^3_3 = \{1\}$. Young tableaux have indices $\text{ind}[3] = 0, \text{ind}[2,1] \in \{1,1\}$, $\text{ind}[1^3] = 2$, from which we deduce for instance $c^{[2]\overline{[2]}}_{(2,0)} = 1 = c^{[11]\overline{[11]}}_{(2,0)},$ $c^{[2]\overline{[11]}}_{(2,1/2)} = c^{[11]\overline{[2]}}_{(2,1/2)} = 1, c^{[3]\overline{[3]}}_{(3,s)} = \delta_{s,0}, c^{[3]\overline{[21]}}_{(3,s)} = \delta_{s,1/3} + \delta_{s,2/3}, c^{[21]\overline{[21]}}_{(3,s)} = 2\delta_{s,0} + \delta_{s,1/3} + \delta_{s,2/3}.$

- As a last example let us compute $c^{[3]\overline{[21]}}_{(4,0)}$. Since $[3]$ has index 0, and $[21]$ has two tableaux with index 1, we are only interested in orbits of $\mathcal{O}^3_4$ of length 4, of which there are only two:

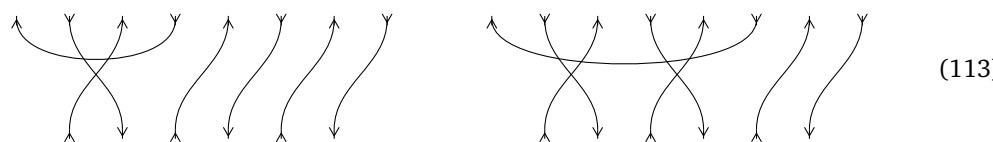

$$\tag{113}$$

This gives $c^{[3]\overline{[21]}}_{(4,0)} = 2 \sum_{\omega \in \mathcal{O}^3_4} \delta_{e^{2i\pi\omega/4}, 1} = 4.$

### 4.3 Comparison with the spectrum of the $O(n)$ CFT

**Spectrum of the $O(n)$ CFT**

The spectrum of the $O(n)$ CFT was guessed in [16], and then derived in [17] using the same techniques that we applied to the $PSU(n)$ case:

$$\mathcal{S}^{O(n)} = \sum_{s\in 2\mathbb{N}+1} \mathcal{R}_{\langle 1,s\rangle} \otimes [\ ] + \sum_{(r,s)\in\frac{1}{2}\mathbb{N}^*\times\frac{1}{r}\mathbb{Z}} \mathcal{W}_{(r,s)} \otimes \Lambda_{(r,s)}, \tag{114}$$

where the $\Lambda_{(r,s)}$ are representations of $O(n)$ defined as

$$\Lambda_{(r,s)} = \delta_{r,1}\delta_{s\in 2\mathbb{Z}+1}[\ ] + \frac{1}{2r}\sum_{r'=0}^{2r-1} e^{\pi i r's}\tilde{T}_{\frac{2r}{2r\wedge r'}}\left(\delta_{2r\wedge r'\in 2\mathbb{N}}[\ ] + \sum_{k=0}^{2r\wedge r'-1}(-1)^k[2r\wedge r'-k,1^k]\right) \tag{115}$$

$$= \sum_{\substack{|\lambda|\leq 2r\\|\lambda|\equiv 2r \bmod 2}} \sum_{T\in T_\lambda} \sum_{\omega\in\Omega^{(2r)}_{|\lambda|}} \delta_{e^{\pi i\omega\left(s-\frac{\mathrm{ind}(T)}{r}\right)},1}\lambda. \tag{116}$$

**Comparison**

The similarity between (114) and (66) is quite striking; on the conformal side, the same kinds of representations appear. We note two differences:

- Non-diagonal fields with half-integer Kac index $r$ only appear in the spectrum of the $O(n)$ CFT.

- Degenerate fields with even Kac index $s$ only appear in the $PSU(n)$ spectrum.

The facts that roughly half of the spectrum of the $O(n)$ CFT is absent in the $PSU(n)$ CFT, and that half of the degenerate sector of the $PSU(n)$ model disappears in the $O(n)$ model, suggest that the $O(n)$ CFT is a $\mathbb{Z}_2$ orbifold of the $PSU(n)$ CFT.

## 5 Orbifold of the $PSU(n)$ CFT

The orthogonal group $O(n)$ is the subgroup of $U(n)$ of matrices invariant under the complex conjugation (11):

$$O(n) = U(n)^c. \tag{117}$$

Can the $O(n)$ model be obtained from the $PSU(n)$ model by modding out conjugation symmetry? While we are unable to answer this question for the lattice models, we will explore it at the level of the corresponding CFTs, where the question becomes: Is the critical $O(n)$ model a $\mathbb{Z}_2$ orbifold of the critical $PSU(n)$ model?

### 5.1 Orbifolds in CFT

In mathematics, a geometric orbifold is the quotient of a manifold by a finite group. Geometric orbifolds are similar to manifolds, except that they are singular at the fixed points of the group action. For instance, the action of $\mathbb{Z}_2$ on the circle $S^1$ by $x \mapsto -x$ has two fixed point at $x = 0$ and $x = \pi$, corresponding to the edges of the orbifold $S^1/\mathbb{Z}_2 = [0,\pi]$.

Given a CFT with a discrete symmetry group $G$, the orbifold CFT is a CFT with the symmetry modded out. The orbifold procedure has a clear definition in the Lagrangian or path-integral formalism. However, the spectrum of the model, which is our main object of study, belongs to the Hamiltonian formalism. We will therefore have to work with an incomplete description of the orbifold procedure in the Hamiltonian formalism, deduced from its definition in the Lagrangian formalism.

**Lagrangian formalism**

Consider a sigma model on a torus $T^2$ of modular parameter $\tau$, with a target manifold $\mathcal{M}$. The partition function is of the type

$$Z(\mathcal{F}(T^2, \mathcal{M})) = \int_{\mathcal{F}(T^2, \mathcal{M})} \mathcal{D}\phi \, \exp(-S[\phi]), \tag{118}$$

where $\mathcal{F}(T^2, \mathcal{M})$ is a suitable space of functions $T^2 \to \mathcal{M}$, and $S[\phi]$ an action functional.

If now a group $G$ acts on $\mathcal{M}$, such that the action is invariant: $S[\phi] = S[g \cdot \phi]$ for any $g \in G$, we define the orbifold CFT as the sigma model with target manifold $\mathcal{M}/G$. Let us write the functions $T^2 \to \mathcal{M}/G$ in terms of functions $\widetilde{T^2} \to \mathcal{M}$, where $\widetilde{T^2} = \mathbb{R}^2$ is the universal covering of the torus:

$$\mathcal{F}(T^2, \mathcal{M}/G) = \sum_{g,h \in G} \mathcal{F}_{g,h}(\widetilde{T^2}, \mathcal{M}) \bigg/ G, \tag{119}$$

where we define the subspaces

$$\mathcal{F}_{g,h}(\widetilde{T^2}, \mathcal{M}) = \left\{ \phi : \widetilde{T^2} \to \mathcal{M} \, \middle| \, \begin{array}{l} \phi(z+1) = g \cdot \phi(z) \\ \phi(z+\tau) = h \cdot \phi(z) \end{array} \right\}, \tag{120}$$

and we mod out by the natural action of $G$ on $\mathcal{F}(\widetilde{T^2}, \mathcal{M})$. The space $\mathcal{F}_{g,h}$ is in fact only well-defined if $gh = hg$, due to the condition $\phi(z+1+\tau) = hg\phi(z) = gh\phi(z)$ [36], but we are interested in the abelian group $G = \mathbb{Z}_2$, so we neglect this subtlety. Thinking of the torus as an open rectangle, we can also view $\mathcal{F}_{g,h}(\widetilde{T^2}, \mathcal{M})$ as a space of functions that obey $G$-twisted boundary conditions.

In these notations, the partition functions of the original sigma model and of the orbifold read

$$Z = Z(\mathcal{F}(T^2, \mathcal{M})) = Z_{1,1}, \tag{121}$$

$$Z^{\text{orb}} = Z(\mathcal{F}(T^2, \mathcal{M}/G)) = \frac{1}{|G|} \sum_{g,h} Z_{g,h}, \tag{122}$$

where we define the partial partition functions

$$Z_{g,h} = Z(\mathcal{F}_{g,h}(\widetilde{T^2}, \mathcal{M})). \tag{123}$$

**Hamiltonian formalism**

The relation between the Lagrangian and Hamiltonian formalisms is called radial quantization. We view the torus as a cylinder whose boundary circles are glued together:

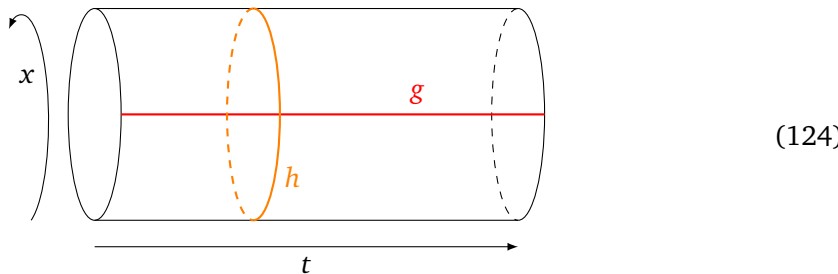

$$\tag{124}$$

The space of states of the theory is the space of field configurations on a space-like slice, which is the set of functions $S^1 \to \mathcal{M}$. In this formalism, the partial partition function $Z_{g,h}$ (123) is interpreted as

$$Z_{g,h} = \text{Tr}_{\mathcal{S}_g} \left( h q^{L_0 - \frac{c}{24}} \bar{q}^{\bar{L}_0 - \frac{c}{24}} \right). \tag{125}$$

Here the element $h \in G$ acts on the $g$-twisted sector of the spectrum,

$$\mathcal{S}_g = \left\{ \phi : \widetilde{S^1} \to \mathcal{M} \,\Big|\, \phi(z+1) = g \cdot \phi(z) \right\}, \tag{126}$$

where $\mathcal{S}_1$ is the untwisted sector. In this formalism, the sum $\frac{1}{|G|} \sum_h h$ in (122) is interpreted as the projection onto $G$-invariant states, and the orbifold partition function is rewritten as

$$Z^{\text{orb}} = \text{Tr}_{\mathcal{S}^{\text{orb}}} \left( q^{L_0 - \frac{c}{24}} \bar{q}^{\bar{L}_0 - \frac{c}{24}} \right), \tag{127}$$

where the orbifold spectrum is

$$\mathcal{S}^{\text{orb}} = (\mathcal{S}^{\text{twist}})^G, \quad \text{with} \quad \mathcal{S}^{\text{twist}} = \bigoplus_g \mathcal{S}_g. \tag{128}$$

This amounts to writing $\mathcal{S}^{\text{orb}}$ as the $G$-invariant subspace of a larger space $\mathcal{S}^{\text{twist}}$ of all states in the untwisted and twisted sectors, respectively $\mathcal{S}_1$ and $\mathcal{S}_{g \neq 1}$. In contrast to $\mathcal{S}$ and $\mathcal{S}^{\text{orb}}$, the larger space $\mathcal{S}^{\text{twist}}$ is not the space of states of a physical model, and does not correspond to a consistent CFT. It is merely introduced as an intermediary object for writing the spectrum of the orbifold theory. In particular, states in $\mathcal{S}^{\text{twist}}$ need not have integer spins $\Delta - \bar{\Delta}$.

By the operator-state correspondence, when a field $V_h \in \mathcal{S}_g$ goes once around $V_g$, the group acts on both fields as follows:

$$V_g(z)V_h(z + e^{2i\pi}w) = \left( h \cdot V_g(z) \right)\left( g \cdot V_h(z+w) \right). \tag{129}$$

In order to account for this non-trivial monodromy, we can represent a twist field $V_{g \neq 1}$ with a trailing topological line going to infinity and carrying a group element $g$:

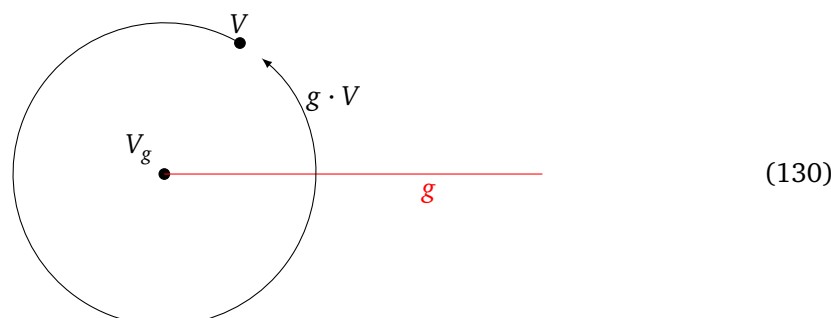

$$\tag{130}$$

In more general settings where the theory is not described as a sigma model, we generalise the definition of an orbifold as follows. Assume a group $G$ is a symmetry of the CFT, meaning that

1. $G$ acts on the space of states $\mathcal{S}$ of the CFT and commutes with the Virasoro algebra.

2. The action of $G$ on operators is compatible with OPEs:

$$V_a V_b \sim \sum_c C_{ab}^c V_c \implies (g \cdot V)(g \cdot V) \sim \sum_c C_{ab}^c (g \cdot V_c). \tag{131}$$

We then define the spectrum of the orbifold CFT by Eq. (128). The difficulty lies in defining the twisted sectors $\mathcal{S}_{g \neq 1}$, for which we know no general procedure.

## 5.2  Conformal symmetry

Let us reproduce the spectra of the $O(n)$ and $PSU(n)$ models from (66) and (114):

$$\mathcal{S}^{PSU(n)} \underset{\mathfrak{C} \times U(n)}{=} \sum_{s \in \mathbb{N}^*} \mathcal{R}_{\langle 1,s \rangle} \otimes [\,] + \sum_{r \in \mathbb{N}^*} \sum_{s \in \frac{1}{r}\mathbb{Z}} \mathcal{W}_{(r,s)} \otimes \Omega_{(r,s)}, \tag{132}$$

$$\mathcal{S}^{O(n)} \underset{\mathfrak{C}_c \times O(n)}{=} \sum_{s \in 2\mathbb{N}+1} \mathcal{R}_{\langle 1,s \rangle} \otimes [\,] + \sum_{r \in \frac{1}{2}\mathbb{N}^*} \sum_{s \in \frac{1}{r}\mathbb{Z}} \mathcal{W}_{(r,s)} \otimes \Lambda_{(r,s)}. \tag{133}$$

We assume that $\mathcal{S}^{O(n)}$ is deduced from $\mathcal{S}^{PSU(n)}$ by a $\mathbb{Z}_2 = \{1, g\}$ orbifold, such that the untwisted and twisted sectors are respectively

$$(\mathcal{S}_1)^{\mathbb{Z}_2} = \left(\mathcal{S}^{PSU(n)}\right)^{\mathbb{Z}_2} = \mathcal{S}^{O(n)}\big|_{r \in \mathbb{N}}, \tag{134}$$

$$\left(\mathcal{S}_g\right)^{\mathbb{Z}_2} = \mathcal{S}^{O(n)}\big|_{r \in \mathbb{N}+\frac{1}{2}}, \tag{135}$$

where by convention $r = 0$ in the degenerate sector. Which $\mathbb{Z}_2$ symmetry of the $PSU(n)$ model could lead to such relations? We start by investigating how this $\mathbb{Z}_2$ should act on the conformal representations $\mathcal{R}_{\langle 1,s \rangle}, \mathcal{W}_{(r,s)}$. This action should be compatible with the conformal fusion rules [37]:

$$V_{\langle 1,2 \rangle}^d \times V_{\langle 1,s \rangle}^d = V_{\langle 1,s-1 \rangle}^d + V_{\langle 1,s+1 \rangle}^d, \tag{136}$$

$$V_{\langle 1,2 \rangle}^d \times V_{(r,s)} = V_{(r,s-1)} + V_{(r,s+1)}. \tag{137}$$

### Degenerate sector and degenerate fusion

Since the degenerate primary fields $V_{\langle 1,s \rangle}^d, s \in 2\mathbb{N}$ are absent from the $O(n)$ model's spectrum, these fields must be odd under $\mathbb{Z}_2$:

$$g \cdot V_{\langle 1,s \rangle}^d = (-1)^{s+1} V_{\langle 1,s \rangle}^d. \tag{138}$$

This action is indeed a symmetry of the fusion rules (136). While the fusion (137) does not determine the sign of $V_{(r,s)}$ under the $\mathbb{Z}_2$ action, it imposes

$$g \cdot V_{(r,s)} = V_{(r,s)} \iff g \cdot V_{(r,s+1)} = -V_{(r,s+1)}, \tag{139}$$

i.e. $V_{(r,s)}$ and $V_{(r,s+1)}$ transform with opposite signs under the $\mathbb{Z}_2$ action.

### Monodromies of $V_{\langle 1,2 \rangle}^d$

We want to interpret the non-diagonal fields $V_{(r,s)}$ as belonging to the twisted or untwisted sector, depending on $2r$ mod 2. According to Eq. (129), this implies in particular that the monodromy of $V_{\langle 1,2 \rangle}^d$ around $V_{(r,s)}$ should be $(-1)^{2r}$. Let us check that this is the case.

Conformal invariance fixes the $z$-dependence of the coefficients in the OPE of two fields $V_1, V_2$ of conformal dimensions $(\Delta_i, \bar{\Delta}_i), i = 1, 2$:

$$V_1(z) V_2(w) \sim \sum_{V_3 \in \mathcal{S}} C_{123}(z-w)^{\Delta_3 - \Delta_1 - \Delta_2}(\bar{z}-\bar{w})^{\bar{\Delta}_3 - \bar{\Delta}_1 - \bar{\Delta}_2} V_3(w). \tag{140}$$

Therefore, the monodromy of $V_1$ around $V_2$ is $e^{2i\pi(S_3 - S_1 - S_2)}$ where $S_i = \Delta_i - \bar{\Delta}_i$ is the conformal spin. Since $V_{\langle 1,2 \rangle}^d$ is diagonal, its conformal spin is zero. The spin of $V_{(r,s)}$ is $rs$, hence the fusion (137) implies that the monodromy of $V_{\langle 1,2 \rangle}$ around $V_{(r,s)}$ is indeed $e^{2i\pi(r(s\pm 1) - rs)} = (-1)^{2r}$.

## 5.3 Global symmetry

As we have just seen, the idea that the $O(n)$ model is a $\mathbb{Z}_2$ orbifold of the $PSU(n)$ model is compatible with conformal symmetry. Let us now see how well the idea fares from the point of view of global symmetry.

**Necessary condition on the spectra**

Given the conjugation $g \mapsto \bar{g}$ on $U(n)$ (11), a conjugation on an irreducible $U(n)$ representation $R$ is an involutive intertwiner $c : R \to \bar{R}$, i.e. a map that obeys

$$c^2 = 1, \qquad \forall g \in U(n), \qquad c\rho(g) = \rho(\bar{g})c. \tag{141}$$

This determines $c$ only up to a sign flip $c \to \sigma_R c$, with a sign $\sigma_R \in \{1, -1\}$ for each irreducible representation. And several copies of the same irreducible representation may come with different signs.

This leaves a lot of freedom on how complex conjugation can act on the $PSU(n)$ spectrum. Nevertheless, let us show that the orbifold relation for the untwisted sector (134) implies a nontrivial relation between the $O(n)$ and $PSU(n)$ spectra. Let us decompose $\Omega_{(r,s)} = \Omega_{(r,s)}^+ + \Omega_{(r,s)}^-$ into eigenspaces of the nontrivial $\mathbb{Z}_2$ element $g$. Since $g$ anticommutes with $s \to s+1$ (70a) while $\Omega_{(r,s)} = \Omega_{(r,s+1)}$ (139), the orbifold relation implies

$$\left(\mathscr{W}_{(r,s)} \otimes \Omega_{(r,s)} + \mathscr{W}_{(r,s+1)} \otimes \Omega_{(r,s+1)}\right)^{\mathbb{Z}_2} = \mathscr{W}_{(r,s)} \otimes \Omega_{(r,s)}^\epsilon + \mathscr{W}_{(r,s+1)} \otimes \Omega_{(r,s)}^{-\epsilon} \tag{142}$$

$$= \mathscr{W}_{(r,s)} \otimes \Lambda_{(r,s)} + \mathscr{W}_{(r,s+1)} \otimes \Lambda_{(r,s+1)}, \tag{143}$$

where $\epsilon \in \{+, -\}$ is defined by $g\mathscr{W}_{(r,s)} = \epsilon \mathscr{W}_{(r,s)}$. Therefore, our two eigenspaces must coincide with the two $O(n)$ representations that appear in the spectrum of the $O(n)$ model:

$$\left\{\Omega_{(r,s)}^+, \Omega_{(r,s)}^-\right\} = \left\{\Lambda_{(r,s)}, \Lambda_{(r,s+1)}\right\}. \tag{144}$$

In particular, this determines how the $U(n)$ representation $\Omega_{(r,s)}$ decomposes into representations of $O(n)$:

$$\boxed{\Omega_{(r,s)} \underset{O(n)}{=} \Lambda_{(r,s)} + \Lambda_{(r,s+1)}.} \tag{145}$$

We will now provide two different proofs that this necessary condition is fulfilled.

**$\Omega = \Lambda + \Lambda$ from the torus partition functions**

The relation (115) implies that the character of the representation $\Lambda_{(r,s)}$ is:

$$\mathrm{Tr}_{\Lambda_{(r,s)}}(g) = \delta_{r,1}\delta_{s\in 2\mathbb{Z}+1} + \frac{1}{2r}\sum_{r'=0}^{2r-1} e^{i\pi r's}\tilde{T}_{(2r)\wedge r'}\left(\mathrm{Tr}_{[1]} g^{\frac{2r}{(2r)\wedge r'}}\right), \tag{146}$$

where the $\tilde{T}_d$ are Chebyshev polynomials introduced in (69). This leads to

$$\mathrm{Tr}_{\Lambda_{(r,s)}+\Lambda_{(r,s+1)}}(g) = \delta_{r,1} + \frac{1}{2r}\sum_{r'=0}^{2r-1} e^{i\pi r's}(1 + e^{i\pi r'})\tilde{T}_{(2r)\wedge r'}\left(\mathrm{Tr}_{[1]} g^{\frac{2r}{(2r)\wedge r'}}\right)$$

$$= \delta_{r,1} + \frac{1}{2r}\sum_{r'=0}^{r-1} e^{2i\pi r's}\tilde{T}_{2(r\wedge r')}\left(\mathrm{Tr}_{[1]} g^{\frac{r}{r\wedge r'}}\right). \tag{147}$$

On the other hand, when $g \in O(n)$, $\text{Tr}_{[1] \otimes \overline{[1]}} g = \left(\text{Tr}_{[1]} g\right)^2$: hence, using $\tilde{T}_{2d} = \tilde{T}_d(X^2 - 2)$, the character of $\Omega_{(r,s)}$ from (91) simplifies to

$$\text{Tr}_{\Omega_{(r,s)}}(g \in O(n)) = \delta_{r,1} + \frac{1}{r} \sum_{r'=0}^{r-1} e^{2i\pi r's} \tilde{T}_{2(r \wedge r')} \left(\text{Tr}_{[1]} g^{\frac{r}{r \wedge r'}}\right), \tag{148}$$

which proves equation (145).

**$\Omega = \Lambda + \Lambda$ from diagram algebras**

Consider the decomposition (101) of the alternating spin chain spectrum (58b). As $O(n)$ representations, the fundamental and antifundamental representations coincide $[1] =_{O(n)} \overline{[1]}$. In categorical terms, there exists a tensorial functor $\underline{\text{Rep}}\, U(n) \rightarrow \underline{\text{Rep}}\, O(n)$ that maps $[1], \overline{[1]} \rightarrow [1]^{O(n)}$. Thus, under $O(n)$ symmetry, the $U(n)$ spin chain reduces to the $O(n)$ spin chain, whose space of states decomposes as [17, eq. (4.15)]

$$\mathcal{S}_{2L}^{O(n)} \underset{u\mathscr{ITL}_{2L}(n) \times O(n)}{=} \sum_{\substack{r \leq L \\ s \in (\frac{1}{r}\mathbb{Z})/2\mathbb{Z}}} W_{(r,s)}^{2L} \otimes \Lambda_{(r,s)}, \tag{149}$$

where the unoriented Jones–Temperley–Lieb algebra $u\mathscr{ITL}_{2L}(n)$ is the algebra generated by $e_i, i = 1, \ldots, 2L$ as in (95) and $u$ is the operator of translation by one site. In order to compare to (101), we need to compute branching rules $u\mathscr{ITL}_{2L}(n) \downarrow \mathscr{ITL}_L(n)$.

Modules of the $u\mathscr{ITL}_{2L}(n)$ algebra are indexed by $r,s$ where $r \in \mathbb{N}^*, s \in (\frac{1}{r}\mathbb{Z})/2\mathbb{Z}$, and they are defined by the action of the pseudo-translation $t$ as

$$t\tilde{W}_{(r,s)}^{2L} = e^{i\pi s}\tilde{W}_{(r,s)}^{2L}. \tag{150}$$

Comparing to (99) we immediately find the following branching rules; for $0 \leq s < 1$:

$$\tilde{W}_{(r,s)}^{2L} \underset{\mathscr{ITL}_L(n)}{=} \tilde{W}_{(r,s+1)} \underset{\mathscr{ITL}_L(n)}{=} W_{(r,s)}^{L}. \tag{151}$$

Plugging these branching rules into eq. (149) yields eq. (145).

# 6 Afterword

**Summary table**

The following table summarizes the relations between the models that we have studied. The first line displays the spectra of the spin chains, and the spaces of configurations of the corresponding loop models. The red box summarizes the algebras that describe interactions between spins. The last line shows the relations between the corresponding CFTs.

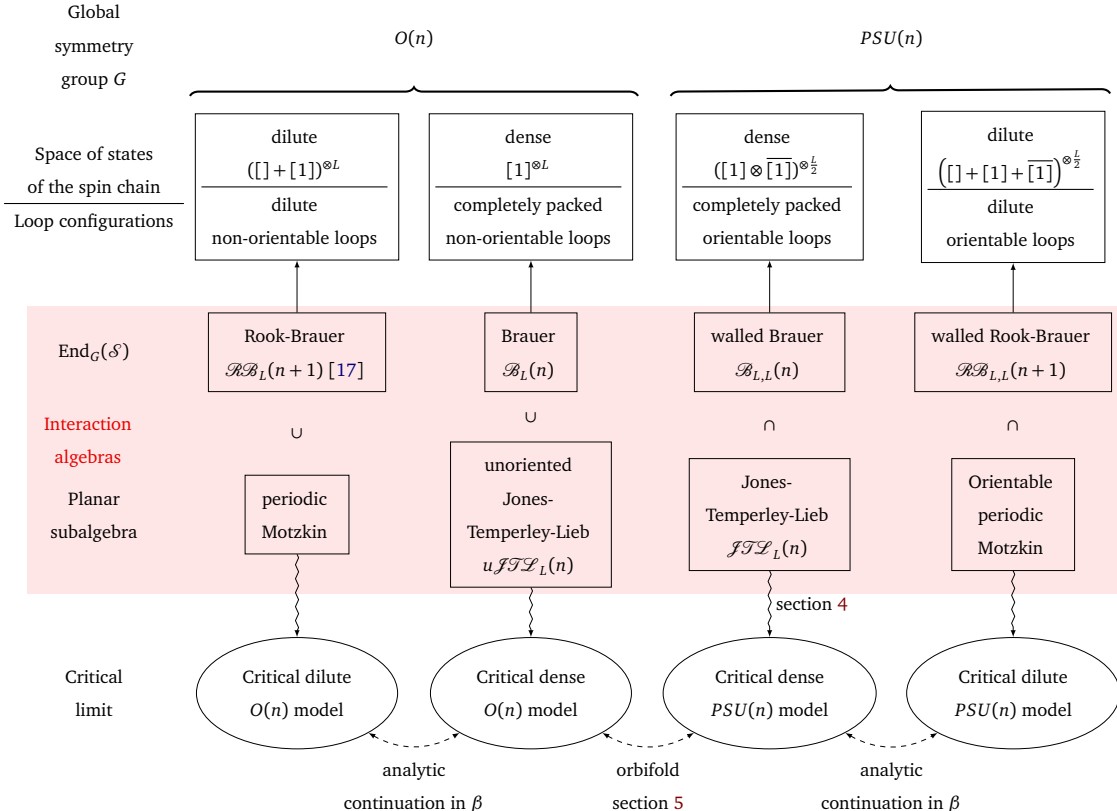

## Further thoughts on the relation between the $O(n)$ and $PSU(n)$ models

While the $O(n)$ and $PSU(n)$ loop models differ only by the orientability of the loops, this leads to significant differences in the corresponding CFTs. In particular, it is instructive to consider the non-diagonal primary fields $V_{(1,-1)}$, which we may be tempted to call currents, because they have left and right dimensions $(\Delta, \bar{\Delta}) = (1, 0)$. According to the $PSU(n)$ model's spectrum (71a), $V_{(1,-1)}$ transforms in the irreducible part of the adjoint representation $[1]\overline{[1]}$ of $PSU(n)$, whose dimension is $n^2 - 1$. In the $O(n)$ model, $V_{(1,-1)}$ transforms in the adjoint representation $[11]$ of $O(n)$, whose dimension is $\frac{n(n-1)}{2}$. These dimensions, in other words these numbers of currents, are a simple way of detecting the CFTs' global symmetries. This invalidates arguments that appeal to $PSU(n)$ symmetry for justifying the genericity of the low-temperature phase of the $O(n)$ model [10].

We have made an algebraic argument that the $O(n)$ CFT is a $\mathbb{Z}_2$ orbifold of the $PSU(n)$ CFT. Let us sketch another argument using a Lagrangian approach. In the same way as the XXX spin chain can be described at low energy by a $\mathbb{CP}^1$ sigma model [38, 39], it has been speculated that alternating $PSU(n)$ spin chains can be described at low energy by a $\mathbb{CP}^{n-1}$ sigma model, see [40] for a thorough review. The $\mathbb{CP}^{n-1}$ sigma model is defined by the Lagrangian

$$\mathcal{L}_\theta = \frac{1}{g_\sigma^2}(\partial_\mu - ia_\mu)Z^\dagger(\partial^\mu + ia^\mu)Z + \frac{\theta}{2\pi}\epsilon^{\mu\nu}\left(\partial_\mu a_\nu - \partial_\nu a_\mu\right), \tag{152}$$

where $Z \in \mathbb{C}^n$, $Z^\dagger Z = 1$, and $a_\mu = \frac{i}{2}\left[Z^\dagger\partial_\mu Z - Z\partial_\mu Z^\dagger\right]$. The field $Z$ should be thought of as transforming in the fundamental representation $[1]$ of $U(n)$, with $Z^\dagger$ transforming in $\overline{[1]}$. Under $Z \to Z^\dagger$, we have $a_\mu \to -a_\mu$ and therefore $\mathcal{L}_\theta \to \mathcal{L}_{-\theta}$. The second term in (152) is therefore not invariant unless $\theta = \pi$, since $\theta$ is defined modulo $2\pi$. Therefore, the $\mathbb{CP}^{n-1}$ sigma model at $\theta = \pi$ has a $\mathbb{Z}_2$ charge conjugation symmetry. At this point, it experiences a first order phase transition for $n > 2$, but it is expected to become gapless for $-2 \leq n \leq 2$, and

flow to a CFT which should be identified with the $PSU(n)$ CFT described in this paper. For integer $n$, the Lagrangian (152) has a $U(n)$ symmetry with a gauged $U(1)$, in particular it has a $PSU(n)$ global symmetry.

In the language of spin chains, the topological angle $\theta$ is coupled to a staggering interaction. At $\theta = \pi$, this staggering disappears, and the spin chain becomes homogeneous, and acquires a symmetry that acts as $[1] \rightarrow \overline{[1]}$ together with translation by one site. It follows that the charge conjugation symmetry of the sigma model should be identified with the $\mathbb{Z}_2$ that is involved in our orbifold. Moving away from $\theta = \pi$ would then correspond to perturbing the fixed point by $V^d_{\langle 1,2 \rangle}$ i.e. modifying the second term in (152). On the other hand, a perturbation by $V^d_{\langle 1,3 \rangle}$ would correspond to changing the coupling, i.e. the first term in (152).

This picture suggests that the $\mathbb{CP}^{n-1}$ model has a more complicated phase diagram when $n \in [-2, 2]$ than for $n > 2$ integer, since both the dilute and the dense critical points for the $PSU(n)$ model are naturally associated with $\mathbb{CP}^{n-1}$ [4]: perhaps a complete picture would require the introduction of additional interaction terms in (152).

In the special case $n = 2$, the $PSU(n)$ and $O(n)$ models simplify, and our orbifold conjecture becomes easy to check. To begin with, the $PSU(2)$ CFT coincides with the $SU(2)$ WZW model at level $k = 1$, and also with a compactified free boson at the self-dual radius $r = 1$ [11]. This coincidence can be seen at the level of torus partition functions:

$$Z^{PSU(2)} = Z^{\text{boson}}(1),\qquad (153)$$

where $Z^{\text{boson}}(r) = Z^{\text{boson}}(\frac{1}{r})$ is the free boson partition function at radius $r$. Similarly, the $O(2)$ CFT has the partition function

$$Z^{O(2)} = Z^{\text{boson}}\left(\frac{1}{2}\right).\qquad (154)$$

However, the free boson at radius $\frac{1}{2}$ is known to be a $\mathbb{Z}_2$ orbifold of the free boson at radius 1 [36]. Moreover, a detailed analysis of the mapping of the $\mathbb{CP}^1$ sigma model to the free boson theory shows that the charge conjugation symmetry in $\mathbb{CP}^1$ corresponds to the $\mathbb{Z}_2$ that is involved in the free boson orbifold.

## Acknowledgments

The authors thank Raoul Santachiara for his comments on this paper. P.R. thanks Mike Zabrocki for his help with the SageMath code for computing the decomposition of tensor products of irreducible representations of $U(n)$, and Mathieu Beauvillain for careful comments as a non-specialist. The authors also thank the three reviewers of SciPost, Bernard Nienhuis and two anonymous, and the editor Slava Rychkov, for a detailed review which has led to important clarifications, in particular on the matter of the global symmetry of our model.

**Funding information** This work is partly a result of the project ReNewQuantum, which received funding from the European Research Council. This work was also supported by the French Agence Nationale de la Recherche (ANR) under grant ANR-21-CE40-0003 (project CONFICA).

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
