# Peer review of "Critical spin chains and loop models with $PSU(n)$ symmetry"

_SciPost Physics, doi:SciPost Phys. 18, 033 (2025)_

## Round 1 · Referee Report · Anonymous (Referee 1) · 2024-9-11

Report

This is an interesting paper which presents new results on U(n) spin chains and loop models. I found the paper a bit difficult to read, though, and I have a number of comments and suggestions of stylistic nature that might improve the presentation and help the reader, see the attached report. The paper will eventually be a very valuable contribution to SciPost.

Attachment

Recommendation

Ask for minor revision

---

## Round 1 · Referee Report · Anonymous (Referee 2) · 2024-9-13

Report

The authors build a CFT arising from a loop model. The claim is that this CFT has $U(n)$ symmetry. I see some issues with this claim, related to the number of currents in general and in special cases.

  1. $U(n)$ has $n^2$ generators, therefore we should expect $n^2$ currents in the holomorphic sector, but the partition function (4.1) only shows $n^2-1$ currents. This is claimed to be related to the fact that the adjoint representation of $U(n)$ is irreducible around eq. (2.3). I disagree with the line of reasoning: one way to see we should still expect $n^2$ currents is that the algebra $u(n) = su(n) ⊕ u(1)$. So we expect $n^2-1$ currents from the $su(n)$ and 1 current from the $u(1)$. It appears to me that the authors disregard the latter.

  2. Specifically, for $U(2)$ we should have 4 currents in the holomorphic sector. This is not the case for the partition function at hand, and in (6.2) it's shown that, for $n=2$, the CFT reduces to a $SU(2)$ model, which does not have $U(2)$ symmetry.

  3. For $n=1$, one should recover some $U(1)$ symmetric model, which is known to have have one conserved current in the holomorphic sector (e.g. the compactified free boson at generic radius). However, the multiplicity of the currents from the partition function (4.1) vanishes here. One way out for this particular issue would be that there's some extra operator that has the dimension of a current for $n=1$, but this seems at odds with the philosophy of obtaining the spectrum of this theory by analytic continuation.

The authors should clarify what the actual symmetry of the CFT described by (4.1) actually is, because currently it looks like it's something smaller than $U(n)$.

Recommendation

Ask for major revision

---

## Round 1 · Referee Report · Bernard Nienhuis (Referee 3) · 2024-10-16

Strengths

1) The paper discusses a very natural class of models that require no further motivation. 2) It thoroughly discusses the necessary mathematics. 3) The reference list contains convenient back references to the referring text.

Weaknesses

1) The technical parts are difficult to follow.

Report

The paper Critical Spin models with U(n) symmetry, by Roux, Jacobsen, Ribault and Saleur, addresses the natural question how spin models with U(n) symmetry are similar to or different from those with U(n) symmetry. In my view it is a very useful contribution to the public knowledge about critical phenomena and their relation with symmetry groups and their representations. As I was studying the paper I was very pleased to see that the references have a link back to the pages where they are mentioned. This greatly improves the experience of reading the paper. Perhaps this is used widely, but I did not notice it before.

My recommendation is that the paper be published in SciPost Physics. I trust that the authors will give due attention to the following remarks.

1) In the abstract the third paragraph states that something exists for all complex n. On first reading I took that to be the U(n) symmetry (group), which I believe does not generally exist. Now that I am compiling my notes to a report, I think it is meant that the CFT with U(n) symmetry exists. Perhaps this ambiguity can be removed.

2) Page 4 line 13 from the bottom. "entirely similar" sounds strange, I think a better choice is "globally similar".

3) I do not follow the reasoning excluding the diagram (3.9). I do not see how allowing this diagram as one of the terms in the Hamiltonian results in the redundancy of a site.

4) I am puzzled by the claim just before (3.11a) that the physics of the dilute and the completely packed model is equivalent. I would say that the physics of the dilute models includes that of the completely packed ones.

5) The text following (3.15) is not correct. In the second sentence the word 'fugacities' is better replaced by 'weights'. The models with the weights (3.15) are integrable only for a special value of x. Therefore, the form for the square lattice weights can not be determined by demanding that the model is integrable, and this is certainly nog argued in ref. [27]

6) The references 18, 24, 27, 28, 34 are incomplete. (and in general the doi codes are printed in an unpleasantly small font.)

Recommendation

Publish (surpasses expectations and criteria for this Journal; among top 10%)

---

## Round 2 · Referee Report · Anonymous (Referee 1) · 2024-11-2

Report

The authors have made a substantial effort of clarification and improvement of their paper. I still have a few minors corrections, suggestions or requests. The paper should then be accepted for publication.

Page 6, line 4 : still no reference [18]. Excessive modesty ?

Page 8, first line of (2.8), \oplus would seem more appropriate than \sum

Page 9, bottom, "tableau" (twice) should read "diagram". (They are not yet filled with integers, contrary to sect. 4 below.)

Page 13, one line above (2.36), a typo:"then then"

Page 18, missing figure in (3.17)

Page 27, the wording "major index" doesn't seem to appear in ref [35], so the authors should either define it, or give a proper reference.

Page 31, missing reference before (5.20)

I'm still surprised by the authors' bias on references. Why not give the full reference and publisher of books ? [19]: Springer; [23] Princeton Univ. Pr; [32]: Springer; and of ref [30] J: Phys. A: Math. Theor. 43 142001 ??

Recommendation

Publish (easily meets expectations and criteria for this Journal; among top 50%)

  • validity: -
  • significance: -
  • originality: -
  • clarity: -
  • formatting: -
  • grammar: -

Author:  Paul Roux  on 2024-11-22  [id 4980]

(in reply to Report 1 on 2024-11-02)

Dear reviewer,

Thank you for your careful remarks.

  • Reference [18] was in fact present, the author's names was a clickable link leading to the repo. I have added the written link as well.
  • We chose to stick with regular + instead of \oplus throughout the paper to lighten notations. I think it would be incoherent to use \oplus only for this expression
  • corrected
  • corrected
  • corrected
  • the major index of a tableau is a combinatorial quantity which cannot be defined very quickly. However it is a standard definition which can be easily found with a quick internet search, for instance it is defined in the wikipedia page on Young tableaux. This is why we chose not to redefine it in the paper.
  • corrected
  • I am sorry about the references. I had to modify the .bst file we used and didn't pay attention to the fact that I had to modify both the article and the book classes in there. This is now corrected.

---

## Round 2 · Referee Report · Bernard Nienhuis (Referee 3) · 2024-11-6

Report

I am satisfied with the authors' reply to my questions / remarks, except for point 3, the argument not to the diagrams (3.9) from the Hamiltonian.
I think I follow the text with which the authors reply to my point, but I do not see what it has to do with my question.
The reasoning given in the manuscript to omit the diagrams (3.9) from the Hamiltonian, when applied to the O(n) chain, would forbid the first diagram of (3.7).

Nevertheless, I recommend publication.

Recommendation

Publish (surpasses expectations and criteria for this Journal; among top 10%)

  • validity: -
  • significance: -
  • originality: -
  • clarity: -
  • formatting: -
  • grammar: -

Author:  Paul Roux  on 2024-11-28  [id 5005]

(in reply to Report 2 by Bernard Nienhuis on 2024-11-06)

Dear reviewer,

Thank you for rightly pointing out that the argument was incorrect. In the latest resubmission we have clearly stated that the alternated orientations is a choice and does not follow from U(n) symmetry or the previous discussion.

---

## Round 2 · Referee Report · Anonymous (Referee 2) · 2024-11-19

Report

I thank the authors for the clarification about the model having $PSU(n) $ symmetry, rather than $U(n)$ symmetry, which answered all the questions I had about the model.

I think the only complaint I have is that the paper still reads as if this model has $U(n)$ symmetry, and this is going to be confusing for a reader. For example, both the title and the abstract still explicitly claim $U(n)$ symmetry.

The CFT is referred to as the $U(n)$ CFT throughout the paper. I think this should be changed to $PSU(n)$ for the sake of clarity. The fact that the language of $U(n)$ is the easiest tool to describe the CFT is secondary, given that it would be indeed possible to describe the same exact CFT using the language of $SU(n)$; the name of the CFT should reflect its global symmetry.

After these changes are implemented, I think the paper is ready to be published

Requested changes

  1. Change the claims of $U(n)$ global symmetry in the title and abstract.
  2. Rename the CFT and the model to clearly represent the global symmetry group.

Recommendation

Ask for minor revision

---

## Round 2 · Author Response

The resubmission aims to clarify the question of $U(n)$ versus $SU(n)$ symmetry in the spin chain, the loop model and the CFT, raised by the reviewers. In short, our model has $PSU(n) = U(n)/U(1)$ symmetry, because $U(1)$ acts trivially on our spectrum. In our particular case, it is possible to make sense of PSU(n) symmetry at generic $n$, because only a particular set of representations of SU(n) appear in the spectrum of the CFT. It is however easier to write the spectrum in terms of representations of $U(n)$, on which $U(1)$ acts trivially.

Below is a copy of the more detailed answer which we gave in a note attached as an answer to the reviews.

$U(n)$ versus $SU(n)$ global symmetry

In our spectrum for the $U(n)$ CFT, there are $n^2-1$ currents, which is the dimension of $PSU(n) = U(n) / U(1)$.

In the alternated spin chain $([1] \otimes \overline{[1]})^{\otimes L}$, while $U(n)$ commutes with the hamiltonian, the $U(1) \subset U(n)$ acts trivially. To call a group a symmetry of a theory all elements of the group need to act non-trivially, except for the identity element. Otherwise, one could let an arbitrarily large group act trivially on the theory and declare that it is a symmetry. Hence it is incorrect to say that the spin chain has $U(n)$ global symmetry, it exactly has $PSU(n)$ global symmetry.

This appears to be a problem since it is noted in [@br19] that no Deligne category exists for the group $SU(n)$, and the known constructions cannot be simply extended to construct one. However, in our spectrum [@rjrs24 eq. (4.1-4.3)], all of the irreducible representations of $SU(n)$ that appear can be represented by Young diagrams of length a multiple of $n$. This is not evident from [@rjrs24 eq (4.6)] since this adopts a $U(n)$-notation for these representations, but is shown in the new equation (2.21), in the new paragraph summarizing the relation between $U(n)$ and $SU(n)$ representations. As we explain in the new paragraph "The category $\underline{\operatorname{Rep}}_0 SU(n)$" at the end of section 2, this particular subset of all representations of $SU(n)$ does admit a Deligne category. This is because these representations admit a canonical bijective mapping to a family of representations of $U(n)$ on which $U(1)$ acts trivially, which we call $\underline{\operatorname{Rep}}_0 U(n)$. Since these representations are closed under tensor product, they have an associated subcategory in the Deligne category $\underline{\operatorname{Rep}}_0 U(n)$, which allows us to define $\underline{\operatorname{Rep}}_0 SU(n)$.

There remains the question of whether we should change the name of the model to the $SU(n)$ model.

If we were to write the representations in eq. (4.6) in terms of representations of $SU(n)$ instead of $U(n)$, the reasonably simple equations (4.6) would involve many complicated diagrams all depending on $n$. We prefer to keep the notation of $U(n)$, keeping in the back of our minds that $U(1)$ acts trivially and so everything could be equivalently written in terms of representations of $SU(n)$ if it was needed. For that reason we chose to keep the name $U(n)$ model.

---

## Round 2 · List of Changes

Answer to the first anonymous referee's questions and remarks

Section 2.

  • See the new paragraph below \"Phase diagrams\"

  • The ranges of summations are infinite, however the coefficients are zero for large diagrams.

  • Defects is the standard term in the theory of diagram algebras. There is a mapping between loop model and RSOS (height) models. In this case inserting defects causes the height field to have a non-trivial monodromy, which is what one expects with QFT defects. However in our case we don't think of the defects as QFT defects.

  • we added a reference for the walled-Brauer algebra, and a definition for Specht modules. Note that the paragraph actually defines the walled-Brauer algebra, no prior knowledge is assumed. we gave details for (2.35) (former (2.26)).

Section 4.

  • we added a definition of the conformal algebra in the intro of the paper and in the beginning of the section, also for $r \wedge 0$.

  • $\omega_1 = [1]$ is true, but in your computation you used $[1] \otimes \overline{[1]} = [1]\overline{[1]}$ while $[1]\otimes \overline{[1]} = [1]\overline{[1]} + [\,]$, see (2.10b).

  • we detailed the comment on the logarithmic structure.

  • factor of $\frac12$ corrected.

  • our expressions are more convenient and more naturally written in terms of the polynomials $\tilde T = 2T(X/2)$ where $T$ are the usual Chebyshev polynomials of the first kind. We changed the notation from $U$ to $\tilde T$ to avoid confusion with polynomials of the second kind.

  • we added a diagram for $u^2$.

  • The interchiral algebra was introduced in [@js18], and we don't want to say much about it here. We simply need to know that it is the algebra generated by the modes of the stress tensor and the degenerate field $V_{\langle 1, 2 \rangle}$ and consequently its modules include all shifts of the $s$ index, as we explain in (4.37 - 4.38).

  • $e_1$ acting on the 7th diagram indeed gives the first diagram. What we meant to say is that a vector is in $W^r_{(r, s)}$ iff it is in $\bigcap_{i=1}^L \operatorname{ker} e_i$, which is a complementary space to $\bigoplus \operatorname{Im}e_i$, because the $e_i$ are (unnormalised) projectors. The space $\bigoplus \operatorname{Im}e_i$ is generated by the first 6 diagrams, hence the complementary has the same dimension as the number of the other diagrams. The resulting representation can be obtained from the diagrams in $\ell^{r, r}_{|\lambda|,|\mu|}$ by imposing that the action of $e_i$ is 0 (remembering that these diagrams are actually just labels for a certain combination of diagrams in $d^{r, r}_{|\lambda|,|\mu|}$) which is in the kernel of all $e_i$s.

  • As for the modular transform of $Z(g)$, it should be interpreted as inserting a vertical $g$-twist, i.e. twisting the Hilbert space of the theory by a group element $g$. This is what we explain in section 5. This description is not useful for the computation of the partition function, hence we didn't include it in section 4.

Answer to the second anonymous referee

For more detail, see the discussion in the first section above.

  • This is correct, there is a $U(n)/U(1) = PSU(n)$ symmetry, hence $n^2-1$ currents.

  • In particular for $n=1$ there is no global symmetry since $PSU(1) = {1}$.

  • Indeed for $n=1$ there are no operators with dimension 1 other than $V_{(1, 1)}$ which has multiplicity 0.

Answer to Bernard Nienhuis

  1. we modified this

  2. modified

  3. In a spin chain, if two neighbouring sites are identical and cannot interact between each other, they can equivalently be replaced by a single site, since the hamiltonian on the spin chain with two identical sites is the one on the spin chain with a single site instead, just shifted by $id_{i, i+1}$.

  4. What is meant is that the CFT that describes the dilute and completely packed models is the same. Said otherwise, the critical behaviour of the models is the same.

  5. Indeed the previous paragraph was badly formulated, to the point where it was wrong. We addressed this.

  6. see the new formatting of references.

---

## Round 3 · Author Response

We have changed the title of the paper and the name of the model to reflect its $PSU(n)$ symmetry.
We have also included minor corrections as suggested by the referees.

---

## Round 4 · Author Response

Corrected title and abstract.

---

## Editorial Decision

published